# Molecular Actors of Inflammation and Their Signaling Pathways: Mechanistic Insights from Zebrafish

**DOI:** 10.3390/biology12020153

**Published:** 2023-01-19

**Authors:** Jade Leiba, Resul Özbilgiç, Liz Hernández, Maria Demou, Georges Lutfalla, Laure Yatime, Mai Nguyen-Chi

**Affiliations:** Laboratory of Pathogens and Host Immunity (LPHI) UMR5294, University of Montpellier, CNRS, INSERM, 34095 Montpellier, France

**Keywords:** inflammation, zebrafish, innate immunity, molecular mechanisms, cell signaling

## Abstract

**Simple Summary:**

Inflammation is a central part of the body’s response to harm that can be elicited by microbes, environmental factors, or internal injuries. This complex physiological process has evolved to protect the body and eliminate the threat. To do so, it relies on immune cells and molecular mediators that act in concert to provide protection and tissue repair. If return to homeostasis fails, prolonged inflammation can cause tissue damage and chronic diseases. It is therefore essential to understand in detail the mechanisms at play during the inflammatory response, in order to modulate them in a pathological context. Such studies strongly benefit from in vivo models, as these can capture the complexity of inflammation as a whole. Zebrafish has proven to be a valuable animal model to study innate immune responses as its immune system shows high similarity with the human one and it offers numerous advantages such as its genetics and transparency. We here review the current knowledge on the molecular actors of inflammation in zebrafish, highlighting how the tools developed to study them have helped gain insights into the mechanisms of inflammation. This will allow to design more refined models of inflammation, mimicking human diseases, for drug screening in zebrafish.

**Abstract:**

Inflammation is a hallmark of the physiological response to aggressions. It is orchestrated by a plethora of molecules that detect the danger, signal intracellularly, and activate immune mechanisms to fight the threat. Understanding these processes at a level that allows to modulate their fate in a pathological context strongly relies on in vivo studies, as these can capture the complexity of the whole process and integrate the intricate interplay between the cellular and molecular actors of inflammation. Over the years, zebrafish has proven to be a well-recognized model to study immune responses linked to human physiopathology. We here provide a systematic review of the molecular effectors of inflammation known in this vertebrate and recapitulate their modes of action, as inferred from sterile or infection-based inflammatory models. We present a comprehensive analysis of their sequence, expression, and tissue distribution and summarize the tools that have been developed to study their function. We further highlight how these tools helped gain insights into the mechanisms of immune cell activation, induction, or resolution of inflammation, by uncovering downstream receptors and signaling pathways. These progresses pave the way for more refined models of inflammation, mimicking human diseases and enabling drug development using zebrafish models.

## 1. Introduction

Inflammation is a natural reaction of the body to harmful stimuli such as pathogenic microbes, damaged cells, toxic chemicals, or physical stresses [1,2]. Orchestrated by the innate immune system, it involves a complex network of cellular and molecular effectors that all act in concert to provide protection and prevent irreversible damage [3,4]. At the tissue level, the basic signs of inflammation are (1) redness, caused by the dilatation of small blood vessels at the site of injury; (2) swelling, linked to the accumulation of body fluids in the injured tissue; (3) fever, resulting from local or systemic raise of the body temperature; (4) pain, due to the action of inflammatory mediators; and (5) possible loss of function of the inflamed tissue [1,2].

The inflammatory response is initiated by the detection of molecular signals that are produced either by microbial invaders (pathogen-associated molecular patterns, PAMPs) or by endogenous cells (danger-associated molecular patterns, DAMPs) [5,6]. This very first line of defense relies on multiple components of the innate immune system, both at the cellular and molecular level, in order to insure rapid and efficient integration of the signals. Increase of blood flow, coupled to the generation of chemotactic gradients, allows leukocytes to rapidly migrate and infiltrate the damaged tissue [7,8]. Neutrophils are massively recruited at first, followed by macrophages [9]. Together with resident cells, they release numerous inflammatory mediators through various mechanisms such as neutrophil extracellular traps (NETs), active secretion, passive diffusion, or even necrosis [4,7,9].

Danger sensors include complement proteins circulating in the blood and pattern recognition receptors (PRRs) expressed by cells lining the inflamed tissue or by immune cells migrating to or carried at the site of injury by body fluids [5,10,11,12]. Complement activation, for example, generates the highly pro-inflammatory complement anaphylatoxins that trigger histamine release, increase vasodilation, and regulate adaptive immunity, through G-protein coupled receptor (GPCR) signaling [13]. On the other hand, signal transduction through PRRs elicits downstream signaling cascades that activate transcription factors such as nuclear factor-κB (NFκB), interferon regulatory factor (IRF), or activator protein-1 (AP-1), leading to the upregulation of various genes coding for inflammatory cytokines, chemokines, or interferons [12].

Inflammation is a tightly controlled process. The acute phase is usually brief and lasts only a couple of days, providing just enough time for the immune system to contain and eliminate the threat, without generating excessive tissue damage [14,15]. This control is elegantly exerted through progressive dilution of the chemotactic gradients that condition leukocyte recruitment, combined with the switching off of their survival program, allowing for an efficient depletion of the professional inflammatory cells in the inflamed tissue [16]. In addition, the production of pro-inflammatory molecules is shut down while pro-resolving mediators are upregulated. These concerted actions enable wound healing and tissue repair. Any failure in this control will delay the resolution process, provoking sustained inflammation, and thereby impairing return to homeostasis. While acute inflammation is essential to the immune response, chronic inflammation has no usefulness. Prolonged exposure to inflammatory stimuli will cause irreversible damage and affect the integrity of the tissue, possibly leading to chronic pathologies [15,17].

Numerous human diseases are elicited or exacerbated by inflammation, including cancers, diabetes, neurodegenerative disorders, autoimmune diseases, allergy, or asthma. Several anti-inflammatory drugs are available on the market, but they are unfortunately not devoid of deleterious side effects and their efficacy in the long term is still a bottleneck. The search for more potent and less toxic anti-inflammatory therapeutics is thus a holy grail that requires intimate knowledge of the mechanisms at play during the acute phase of inflammation and during the switch from acute phase to resolution of inflammation. As these processes are highly intricate and involve multiple actors, both at the cellular and molecular level, fundamental and clinical research in the field of inflammation strongly relies on in vivo models that integrate the full complexity of the inflammatory response.

Animal models such as mice and rats are commonly used to decipher immune responses and inflammatory processes in diseases. However, these models are expensive, subjected to ethical consideration, and not adapted to large scale analyses such as drug screening. Alternative translucent organisms have made their entrance in research a long time ago, but with increasing knowledge and availability of tools combined with the rapid progress of technologies such as photon microscopy, they have become popular models for biomedical research. In particular, the zebrafish embryo (Danio rerio) has emerged as a powerful vertebrate model system for understanding mechanisms underpinning inflammation. Its numerous advantages over other species may explain the popularity of the zebrafish. Housing is cheap and not space-consuming, and breeding is fast and yields abundant progeny. Importantly, sequencing of the zebrafish genome has revealed great similarities at the genomic and molecular level with other vertebrates, including humans [18,19]. The amenability of zebrafish to genetic manipulation also allows to use techniques such as transgenesis, protein overexpression, genome editing, or large-scale genome mutagenesis [20]. Furthermore, in the early stages of its development, the zebrafish embryo/larva is permeable to small chemical compounds, allowing large-scale testing of chemical compounds–physiology interactions [21]. Finally, the most advantageous feature of the zebrafish embryo is its transparency, which enables high-resolution imaging within live animals. Populations of cells can be traced with specific color coding thanks to dedicated genetic tools, and cells’ behavior and function can be monitored in vivo and in toto in real time.

Thanks to all these assets, the zebrafish model is currently used in many laboratories as a replacement for studies conducted in mice. As with all animal models, it evidently presents some drawbacks that may hinder proper study of the immune response and inflammatory processes. Some biological analysis tools are, for example, lacking, such as antibodies. Certain genes involved in important pathways are also poorly conserved or simply not existing in the zebrafish genome, suggesting that some molecular functions are either absent or exerted by distantly related, or even unrelated effectors. The development of single-cell -omics approaches and the generation of humanized zebrafish lines have, however, allowed to circumvent some of these disadvantages, making zebrafish a powerful model to study innate immunity and inflammation in connection to human physiopathology [22,23].

Zebrafish have been successfully used to model human diseases in recent years [24], and more particularly inflammatory processes, using either infection models by various microorganisms or models of sterile inflammation relying on physical injury, chemical induction, or mutations [25,26]. While many excellent reviews have extensively described these zebrafish inflammatory models, we here focus on the molecular effectors of inflammation known in zebrafish. We provide an exhaustive list of the inflammatory molecules characterized so far in zebrafish and recapitulate the current knowledge on their modes of action, as inferred from inflammatory models. This review covers major protein and chemical effectors, including cytokines, chemokines, alarmins, complement proteins, proteases, as well as lipids and free radicals. We present a comparative analysis of their sequence, expression, and tissue distribution with regards to their mammalian counterparts and summarize the tools that have been developed in zebrafish to study their function. We further highlight how these tools have helped gain insights into the mechanisms of inflammation in zebrafish, uncovering the receptors and signaling pathways activated downstream of the inflammatory mediators, as well as their effects on immune cell recruitment, activation, and induction or resolution of inflammation. These progresses pave the way for the design of more sophisticated models of inflammation in zebrafish, mimicking human inflammatory diseases and enabling pharmacological development.

## 2. Cellular Components of Inflammation in Zebrafish

In response to any kind of threat, innate immune cells from the local environment, comprising neutrophils and macrophages, get immediately activated and migrate to the site of injury to clear pathogens, dying cells, or cellular debris (Figure 1). As central actors of the inflammatory response, they orchestrate each step of this process, from initiation to resolution [27].

The zebrafish innate immune system is very similar to that of mammals in terms of cellular components and function. As in mammals, zebrafish myeloid cells arise from successive hematopoietic waves. The first macrophages, known as primitive macrophages, emerge between 12 and 24 h post fertilization (hpf), in the yolk sac, and colonize embryonic tissues [28]. They display a phagocytic activity and are capable to activate from 26 hpf [29,30]. Definitive macrophages arise later in the development and are derived from erythromyeloid progenitors and hematopoietic stem cells [31,32]. Zebrafish neutrophils initially develop at 18 hpf and become mature around 35 hpf, displaying features of human neutrophils, such as granules and myeloperoxidase activity [33,34].

Multiple transgenic lines have been developed over the years and are extremely precious to study innate immune cells in zebrafish (Table 1). Tg(*mpeg1:GFP*), Tg(*mfap4:mCherry-F*), and Tg(*mpeg1:mCherry-F*) [35,36,37,38] are commonly used to visualize macrophages whereas Tg(*mpx:eGFP*), Tg(*lysC:DsRed*), and Tg(*lysC:eGFP*) [39,40] allow visualization of neutrophils. Alternatively, Tg(*pu.1:GFP*) and Tg(*coro1a:eGFP*) are also commonly used to track all myeloid cells [41,42]. Valuable tools also exist to deplete leukocytes or to increase innate immune cell number.

While the innate branch of the immune system is fully functional from 2 dpf, the adaptive immunity is only mature after 3 weeks of development (reviewed in [43]), allowing to study the contribution of innate immune cells to inflammation in zebrafish larvae, independently from the adaptive immune system. Like mammals, the zebrafish possesses an adaptive immune system that allows memory responses to pathogens. The zebrafish expresses a number of genes related to adaptive immunity, such as recombinase activating gene (RAG) and T-cell and B-cell receptor (TCR and BCR) genes [44], and it produces immunoglobulins. It also possesses multiple classes of lymphocytes T (including αβ and γδ T cells) [45] and lymphocytes B (B cells) [44]. T cells have been identified in zebrafish and studied mostly using transgenic reporter lines such as Tg(*lck:eGFP*) [46]. Transgenic fluorescent lineage reporters available for studying B cells are Tg(*IgM1:eGFP*) [47] and Tg(*cd79:GFP*) [44]. Innate lymphoid cells, which develop from common lymphoid progenitors, have also been identified in zebrafish and were shown to react to immune challenge by expressing cytokines associated with type 1, 2, and 3 responses [48].

**Table 1 biology-12-00153-t001:** Tools generated to study the cellular components of inflammation in zebrafish, including fluorescent reporter lines, morpholinos, mutants, and systems to deplete or enrich specific immune cell populations. Tg: transgenic line; MO: morpholino; mRNA: mRNA microinjection.

Tool Name and Application	Target Cells	Description	References
Immune cell visualization (transgenic lines)			
Tg(*mpeg1:Gal4-VP16)*, Tg(*mpeg1:GFP-caax)*, Tg(*mpeg1:mCherry-F*), Tg(*mpeg1:GFP)*	Macrophages	Reporter lines, *mpeg1* promoter	[36,37,49]
Tg(*mfap4:tdTomato*), Tg(*mfap4:mCherry-F*)	Macrophages	Reporter lines, *mfap4* promoter	[35,38]
Tg(*tnfα:GFP-F*)	Cells expressing TNFα	Reporter line, *tnfα* promoter	[50]
Tg(*mpeg1:mCherry-F/tnfα:GFP-F*)	M1-like macrophages	All macrophages are labelled with mCherry, pro-inflammatory macrophages are also labelled with GFP	[50]
Tg(*il1β:eGFP-F*), TgBAC(*il1β:eGFP*)	Cells expressing IL1β	Reporter lines, *il1β* promoter	[49,51,52]
Tg(*mfap4:mCherry-F*/*tnfα:GFP-F*)	M1 macrophages	All macrophages are labelled with mCherry, pro-inflammatory macrophages are also labelled with GFP	[53]
Tg(*irg1:eGFP*)	M1 macrophages	Activated macrophages are labelled with GFP	[54]
Tg(*lysC:dsRed*), Tg(*lysC:eGFP*)	Neutrophils	Reporter lines, *lysC* promoter	[40]
Tg(*mpx:eGFP*)	Neutrophils	Reporter line, *mpx* promoter	[39]
Tg(*pu.1:GFP*)	Myeloid cells	Reporter line, *pu.1* promoter	[41]
Tg(*coro1a:eGFP*)	Macrophages, Neutrophils	Reporter line, *coro1a* promoter	[42]
Tg(*lck:eGFP*)	T cells	Reporter line, *lck* promoter	[46]
Tg(*IgM1:eGFP*)	B cells	Reporter line, *IgM1* promoter	[47]
Tg(*cd79:GFP*)	B cells	Reporter line, *cd79* promoter	[44]
Tg(*krt4:nlsEGFP*)	Superficial skin cells	Reporter line, *krt4* promoter	[55]
Immune cell depletion system			
Tg(*mpeg1:Gal4/UAS-nfsb:mCherry*)	Macrophages	Nitroreductase/MTZ treatment	
Tg(*mpx:Gal4/UAS-nfsb:mCherry*)	Neutrophils	Nitroreductase/MTZ treatment	[56]
Tg(*lysC:Gal4/UAS-nfsb:mCherry*)	Neutrophils	Nitroreductase/MTZ treatment	[57]
Clodronate liposomes	Macrophages	Liposome-mediated depletion	[36,58]
MO-*csf3r*	Neutrophils	Morpholino-based depletion	[37,59]
MO-*irf8*	Macrophages	Morpholino-based depletion	[60]
Immune cell enrichment system			
MO-*irf8*	Neutrophils	*irf8* knock-down leads to macrophage depletion and neutrophil enrichment	[60]
mRNA-*irf8*	Macrophages	Macrophage enrichment through overexpression of *irf8*	[60]
mRNA-*gcsfa/b (csf3a/b)*	Neutrophils	Neutrophil enrichment through overexpression of *gcsfa* or *gcsfb*	[61]
*pu.1^G242D^* mutant	Neutrophils	Suppression of *pu.1* triggers neutrophil expansion	[62,63]

The various epithelia of the body also represent crucial protective barriers against intruding pathogens and key regulators of homeostasis [64]. Any damage to these barrier tissues leads to the release of alarm signals by the epithelia that will then produce pro-inflammatory cytokines [65,66]. In contrast to mammals, epithelia of the fish are relatively thin and simple, but mammalian genes involved in epithelial integrity and homeostasis are conserved in zebrafish. In zebrafish larva, the skin epithelium is made of enveloping and basal layers including keratinocytes, mucus cells, ionocytes, and basal cells. This epithelium was shown to release early wound signals working as immediate stress signals for rapid detection of the wound by the immune system [67,68]. Moreover, epithelial cells are the first site of entry for invasive intestinal pathogens, and keratinocytes provide a natural reservoir for many alarmins, such as S100 proteins [69], or for inflammatory cytokines such as pro-IL1β [70,71], thus allowing to initiate the mucosal inflammatory response. Precious tools have been developed in zebrafish to fluorescently tag specific epithelial cells [72] and to perform conditional zebrafish skin ablation [55].

## 3. Molecular Mediators of Inflammation in Zebrafish

At the molecular level, inflammation is induced and relayed by many different types of effectors (Figure 1). Complement proteins, alarmins, and chemical signals act as early sensors of the danger and trigger the first inflammatory events. These are amplified by the production and release of various cytokines and chemokines, and by intracellular activation of inflammasomes. In later steps, protein and lipidic pro-resolving mediators take over to shut down inflammation and promote return to homeostatic conditions. In addition to these central actors, many other proteins contribute to inflammation, more or less directly. These include components of the cytoskeleton such as septins, adhesion molecules that promote cell–cell contacts and motility, or matrix metalloproteases that regulate physical barriers and chemotactic gradients. In this part, we focus on the most recognized molecular mediators of inflammation. We review those that have been identified in zebrafish, compare them to their mammalian counterparts, and list the tools that have been developed to study their function in zebrafish (Table 2).

### 3.1. Small Molecules and Free Radicals

Insults such as injuries trigger the immediate release of damage signals also known as “early wound signals”, whose aim is to initiate the wound signaling cascade to eventually restore the integrity of the epithelial barrier. These small molecules or chemicals have the particularity to diffuse freely in tissues as extracellular paracrine signals or to rapidly propagate from cell to cell. Identified for their capacity to chemoattract leukocytes to the injury site in many species, including zebrafish, the early wound signals are thought to be directly detected by immune cells working in a transcription-independent manner [68]. The composition and magnitude of these signals are tightly regulated and depend on the context. The specific sequence of events resulting in their production, as well as the pathways or crosstalk between them are still debated. Thanks to the use of fluorescent living probes and biosensors, early wound signals can be visualized at high spatio-temporal resolution in the zebrafish larva, shedding light on new molecular and cellular mechanisms orchestrated by these damage signals. The best understood of these are reactive oxygen species (ROS) which play a pivotal role in the regulation of inflammation. Using a genetically encoded fluorescent sensor protein, HyPer, to vizualize hydrogen peroxide (H_2_O_2_), Niethammer et al. showed that within minutes after damage, a tissue gradient of H_2_O_2_ is produced at the wound margin in epithelial cells. This H_2_O_2_ gradient acts as a second messenger signal that drives leukocyte recruitment [113,114].

H_2_O_2_ is unlikely to be the first signal emitted. The first signal at the wound was proposed to be calcium. As a mediator of cell stress, Ca^2+^ signaling shows up in the very first minutes following injury as shown by the use of the genetically encoded calcium fluorescent sensor GCaMP and may be one of the earliest signals [67,79,115]. Ca^2+^ spikes correspond to transient elevations of Ca^2+^ in the cytosol that spread to neighboring cells via cell-to-cell communication (Gap junctions). Alternatively, an extracellular second messenger mediating paracrine signaling can trigger calcium mobilization from internal stores or intake through channels or transporters. In that way, Ca^2+^ rapidly propagates in the tissue from the wounded area and it was shown to participate in the recruitment and activation of innate immune cells in zebrafish [53,67,79,112]. Other early wound signals comprise nucleotides such as ATP, which is thought to be released from dying cells and recognized by purinergic receptors [112] and amino acids such as glutamate [79]. Most of these signals are conserved across species and play similar roles in Drosophila [116], Xenopus [117], and mammals [118], suggesting that they represent ancient danger signals.

### 3.2. Complement

Complement is another ancestral component of immunity. It provides a first-line defense against pathogenic microbes, as well as immune surveillance and clearance, in order to control body homeostasis [119,120]. This heat-labile component of the plasma regroups more than forty proteins, circulating either as soluble factors or as receptors bound to the membrane of immune cells. A more or less complete version of the system is found in most eukaryotic organisms, including teleost fish. Zebrafish complement has been characterized in detail and most mammalian genes possess orthologs in that organism, including the central C3, C4, and C5 genes [121]. As in mammals, complement genes seem to be expressed in the liver, at least for those that have been studied in more detail [121]. Complement components are also acquired through maternal transfer, either at the mRNA or protein level, allowing for efficient protection against harmful stimuli from the earliest stages of development [122].

Complement activation leads to a complex biochemical cascade that ultimately releases the C3a and C5a anaphylatoxins in the blood [13]. These small peptides, together with their cognate G-protein coupled receptors (C3aR, C5aR, and C5L2), are instrumental in triggering the acute inflammatory response [123]. Through chemotaxis, they attract immune cells to the site of injury. They also induce degranulation of mast cells and basophils, which in turn produce histamine and proteases that promote vasodilation, increase of vascular permeability and smooth muscle contraction. These combined events all contribute to local inflammation.

One C5 and three C3 genes have been identified in zebrafish, the latter three resulting from gene duplication [121]. Five additional C3 genes have been reported by the Novoa group [102] but further validation still awaits. Functional C3aR and C5aR are also present in zebrafish [23,103]. Only one gene has been identified for each of these receptors while no gene has been detected for C5L2 so far. To date, very few studies have been performed on complement anaphylatoxins and their receptors in zebrafish, thus limiting the availability of tools to analyze their function. Nevertheless, morpholinos to knock-down C3a or C3aR genes as well as C3a-targeting RNA probes for in situ hybridizations have been reported [102,103]. The activity of zebrafish C5aR has also been tested in cellular models, revealing specificity for the zebrafish C5a anaphylatoxin as compared to the human one and unresponsiveness to human-specific virulence factors [23]. Finally, while the function of zebrafish C5aR has not been addressed directly in vivo, a humanized zebrafish model expressing human C5aR in neutrophils has been generated by the Renshaw group, allowing to study in more detail complement-mediated responses to human pathogens in this vertebrate model [23].

### 3.3. Alarmins

Another set of early triggers of inflammation corresponds to alarmins, a subclass of DAMPs that exert cytokine-like functions [124]. As for all DAMPs, these molecules are produced intracellularly during homeostasis but get released in the extracellular space from damaged or necrotic cells following a sterile or microbial insult. As extracellular components, they perform diverse immune functions such as chemoattraction, differentiation and activation of immune cells, and most importantly, pro-inflammatory signaling through PRRs. A non-exhaustive list of alarmins found in mammals include S100 proteins, histone-like proteins such as High Mobility Group Box 1 protein (HMGB1) and NCAMP-1, heat-shock proteins (HSPs), interleukins 1α (IL1α), and 33 (IL33), and granule-derived proteins and peptides (defensins, granulysin, cathelicidin) [124,125]. The most well-characterized alarmins will be discussed in the following section.

#### 3.3.1. S100 Proteins

S100 proteins are 10–15 kDa calcium-binding peptides, belonging to the EF-hand superfamily [126,127]. Only present in vertebrates, they are expressed in the cytosol of a wide range of cells and may move to the nucleus for DNA repair. Up to twenty-five distinct S100 exist in mammals. They are detected at low levels in most tissues and display a S100-specific enrichment profile [128]. Noteworthily, many of them are present in tissues highly exposed to the environment, such as the skin, gut, respiratory tract, or epithelia, as well as bone marrow and lymphoid tissues [127,129]. Early danger signals stimulate their extracellular release through passive diffusion from injured cells, vesicular secretion within NETs, or other non-conventional secretory mechanisms that are not fully characterized yet.

S100 functions as alarmins include chemotaxis, regulation of macrophage polarization, accumulation of myeloid-derived suppressor cells during tumor growth, and signaling through various cell-surface receptors, such as the receptor for advanced glycation end-products (RAGE), Toll-like receptor 4 (TLR4), basigin, or neuroplastin. S100 proteins have been shown to contribute to inflammation in many pathological settings, including infections, cancers, cardiovascular diseases, and even neurodegeneration [126,130]. In addition, several members of the family are considered as general inflammatory markers [131,132].

Recent phylogenetic studies identified fourteen S100 genes in zebrafish [69]. Among these, six are found in all other vertebrates, pointing out at a conserved subgroup from which the ancestral S100 gene may have arisen [133]. The eight other S100 genes from Danio rerio are only found in teleost fish. Little is known about these zebrafish S100. Expression data obtained from single-cell RNA-sequencing [134], qPCRs on total RNA from adult tissues and in situ hybridization at larval stages have shown that quite a few of these genes are expressed in the zebrafish skin, gut, gills, and various epithelia [69]. Expression has also been detected in macrophages and neutrophils [134]. As studies on zebrafish S100 only start emerging, very few tools have been constructed so far. A S100A10a-specific morpholino and a CRISPR mutant for S100I2 have been reported [104,110]. It is also worth mentioning that several antibodies against mammalian or other fish S100 proteins have shown cross-reactivity in zebrafish.

#### 3.3.2. HMGB1

High-mobility group box proteins (HMGBs) are histone-like DNA-binding proteins that are expressed in the eukaryotic nucleus and participate in chromatin remodeling and transcriptional regulation during homeostasis [135]. HMGB1 is the most studied member of the family as it also displays important immune functions, in particular in its extracellular form. Following tissue injury or infection, HMGB1 is indeed released by necrotic cells, activated immune cells, or as a component of NETs. Extracellularly, it can associate with various cytokines, such as CXCL12 and IL1β, LPS, or CpG oligodeoxynucleotides to induce an inflammatory burst through promiscuous binding to TLRs, RAGE, or CXCR4 [136,137,138,139,140]. HMGB1-dependent activation of human peripheral blood monocytes and macrophages will for example produce IL1, IL6, IL8, and TNFα [141]. HMGB1 also acts as a late mediator of infection-driven inflammation, such as during sepsis, where it is secreted by activated macrophages and monocytes exposed to pathogens in a later time window than other cytokines [142]. In addition, HMGB1 displays immunosuppressive properties that impair proper resolution of inflammation [143].

The inflammatory properties of HMGB1 are highly dependent on its redox state. HMGB1 contains three cysteine residues, the first two of which can associate into an intramolecular disulfide bridge while the third one remains in the thiol state. This disulfide form of HMGB1 is the pro-inflammatory form that elicits the cytokine storm, while fully reduced HMGB1 has chemoattractant properties through CXCR4 signaling but does not induce cytokine release [144]. This fully reduced form also promotes regeneration of injured muscles and liver tissues, as observed in murine models of acute muscle injury or drug-induced liver injury [145]. HMGB1 thus acts as a double-edged sword during inflammation.

The HMGB family has four members in mammals that all show different tissue and cell expression patterns. Six orthologs have been described in zebrafish, including two paralogs of HMGB1 (HMGB1a and 1b) [146]. Morpholinos to knock-down HMGB1a expression in zebrafish have been described [87,88]. Transgenic lines with promoter-driven expression of HMGB1a-GFP or HMGB1a-mCherry fusion proteins in pancreatic β-cells or Notch-responsive cells have also been generated to specifically label the nuclei of these cells [80].

#### 3.3.3. NCAMP-1

Nonspecific cytotoxic cell antimicrobial protein (NCAMP-1) is an H1X-like histone protein first identified in catfish and mouse [147,148] and later on in zebrafish [149]. In mammals, it is expressed in the membrane and in granule extracts from various populations of leukocytes, including natural killer (NK) cells and macrophages [148]. In zebrafish, NCAMP-1 is detected in the coelomic cavity of various tissues, including kidney, liver, and intestine. In these tissues, it is present in the cytosol of both epithelial cells and various mononuclear cells, in particular nonspecific cytotoxic cells (NCCs), the equivalent of mammalian NK cells [149].

NCAMP-1 was shown to be secreted from cultured zebrafish coelomic cells in response to ATP stimulation. Outside cells, it is capable of activating the P2X7 purinergic receptor, inducing its pore formation and subsequent dye uptake. It also triggers intracellular calcium influx [149]. While the NCAMP-1 receptors and downstream signaling pathways remain elusive in zebrafish, the catfish protein has been described as an activator of the caspase-1 inflammatory pathway, capable to initiate IL1β release [150]. All these elements have contributed to categorize NCAMP-1 as a genuine alarmin functioning similarly to HMGB1, ATP, and cathelicidin [149].

#### 3.3.4. HSPs

Heat shock proteins (HSPs) are produced by eukaryotic cells in response to stressful stimuli that include temperature changes (both heat and cold shocks), UV light, physical injuries, and pathogenic microbes. They exert an essential role of chaperones, promoting correct protein folding in conditions where enhanced protein synthesis is required and/or helping refold proteins that were denatured due to cell damage [151]. They are named according to their molecular size, the smallest one, ubiquitin, being 8 kDa big while the largest one, HSP110, weighs more than 100 kDa.

In response to damage, HSPs are secreted via non-classical pathways, including exosomes, or released passively by necrotic cells [152]. Extracellular HSPs can interact with several receptors such as TLRs and scavenger receptors to induce the secretion of pro-inflammatory cytokines such as TNFα, IL1, or IL6 through NFκB pathways [153,154,155]. They also possess a major role in adaptive immunity through facilitation of MHC-antigen processing and presentation [156]. Their role as genuine DAMPs has, however, raised a controversy as several reports prompt them to be rather anti-inflammatory and immunosuppressive [157,158,159]. One way or the other, HSPs are undoubtfully involved in the inflammatory response.

Quite a few HSP genes have been identified in zebrafish, including several members of the HSP47, HSP70, and HSP90 families [160]. The heat shock response has been well studied in zebrafish, together with the role of HSPs during embryonic development [160,161,162,163,164]. Information on the inflammatory functions of zebrafish HSPs are scarcer but extracellular HSP60 has, for example, been shown to act as a leukocyte chemoattractant in hair cell or caudal fin injury models [165]. Various fluorescent reporter lines are available to follow the fate of HSPs in Danio rerio or to obtain heat-inducible transgenic lines, based among others on HSPB1 (HSP27) and HSP70 family members [166,167]. Morphants, CRISPR-Cas9, and TALEN mutants have also been generated for several of these genes.

#### 3.3.5. Cationic Antimicrobial Peptides (CAMPs)

CAMPs form a large class of antimicrobial peptides that includes defensins, cathelicidins, granulysin, and the C-terminal degradation products of chemokines, thrombocidins. These peptides are involved in host defense against bacterial, fungal, and viral infections [168,169,170,171]. They contribute to the antimicrobial activity of various types of leukocytes, as well as keratinocytes, epithelial, and mucosal cells, and facilitate the killing of invading microbes by host cells, presumably through interaction with the microbial cell membrane [172]. Although their precise mechanism of action is still unclear, it involves permeabilization of the pathogen cell membrane by pore formation, operated either directly by the CAMP or by an accessory protein [168,169,170,171]. This mode of action seems to be shared by defensins, cathelicidins, and granulysins, while thrombocidins may exert a microbicidal activity independent of pore formation [171]. Thrombocidins will not be discussed further here as they are derived from chemokines, described later in this review (see Section 3.5).

Three classes of defensins exist in mammals: α, β, and θ-defensins, encompassing more than 100 distinct genes in total [168]. Only α and β subclasses are found in humans. The sole cathelicidin peptide found in human is LL37, but several mammals contain more than one gene for this CAMP family [170]. Granulysin is also represented by a unique gene in humans and other mammals. Aside from their direct antimicrobial action, it is now well-recognized that CAMPs exert immunomodulatory functions that are essential in providing protection against invading microbes, but are also required in non-infectious contexts such as during tumorigenesis [169,173,174]. Most CAMPs display chemotactic properties toward leukocytes. Defensins can attract monocytes, mast cells, immature dendritic cells, and certain subsets of T cells. Cathelicidins recruit neutrophils, monocytes, mast, and T cells, while granulysin is only chemotactic for monocytes. Furthermore, CAMPs can activate immune cells and regulate their expression of pro-inflammatory cytokines and chemokines. α-defensins have, for example, been shown to increase the levels of various cytokines, such as IL5, IL6, IL10, and IFNγ [175]. β-defensins can upregulate the pro-inflammatory TNFα, IL1α, IL6, IL8, CCL2, and CCL18, as well as the anti-inflammatory IL10 [174,176]. Similarly, granulysin enhances the production of IL1, IL6, IL10, IFNα, CCL2, CCL3, CCL5, and CCL7 [177]. Defensins and cathelicidins can also suppress pro-inflammatory signals by downregulating cytokines such as TNFα, IL1, IL6, or IL8 [178,179,180]. CAMPs thereby act as important modulators of inflammation.

To date, three beta-like defensin genes have been characterized in zebrafish [181]. Studies on these teleost defensins have so far focused on their antibacterial and antiviral properties [182,183]. At least two zebrafish-specific cathelicidins were also identified and displayed antimicrobial properties [184]. Finally, four genes orthologous to human granulysin were found in zebrafish, two of which were shown to be upregulated during bacterial infection [185]. However, their participation in bacterial killing or in inflammatory processes remains to be investigated. Genetic tools to deeper analyze the immune properties of zebrafish CAMPs are clearly needed to unravel their immunomodulatory and inflammatory functions in this vertebrate.

### 3.4. Cytokines (Lymphotoxins and Interleukins)

Cytokines are small secreted proteins playing crucial roles in host response and acute inflammation. They include lymphotoxins, interleukins, chemokines, and interferons. The two last subclasses will be described in more detail in dedicated paragraphs (see below). Once released, cytokines can act on the cells that secreted them, as an autocrine signal, or on distinct cells, as a paracrine or endocrine signal. Through binding to specific receptors, they activate a cascade of signaling pathways that have been intensively studied in various pathologies including inflammatory disorders [186]. The main pro-inflammatory cytokines are tumor necrosis factor alpha (TNFα) and interleukin-1 beta (IL1β). TNFα plays key functions in the maintenance of homeostasis and disease pathogenesis, and therapeutic strategies targeting TNFα were shown to provide efficient treatment of inflammatory diseases such as rheumatoid arthritis (RA) and inflammatory bowel diseases (IBD). Activated TNFα exists as a transmembrane form, but, when necessary, it can be cleaved by TNFα-converting enzyme (TACE) and then released as a soluble form. TNFα exerts pleiotropic effects on various cell types through the binding to its receptors, TNFR1 (TNFRSF1A) and TNFR2 (TNFRSF1B) (reviewed in [187]).

IL1β is also an important initiator of inflammation. Its bioactivity results from a sequence of events that is unconventional and tightly regulated. First, in response to danger signals, it is produced as an inactive precursor, named pro-IL1β. Second, as a consequence of inflammasome formation, the protease caspase-1 is activated and cleaves pro-IL1β into an active and secreted form, termed cleaved IL1β. During inflammation, cleaved IL1β signals through IL1 receptor type I (IL1RI), while IL1 receptor type II (IL1RII) was proposed to act as a repressor of IL1β activity [188]. Pro-IL1β can not only be activated through caspase cleavage upon cell stimulation, but it can also be activated by extracellular proteases after wounding and released from injured epithelial cells such as keratinocytes. It has been shown that matrix metalloproteases (MMPs) can convert pro-IL1β to active IL1β [189], but also degrade active IL1β [189], therefore playing a role both in stimulating and then shutting down inflammation.

TNFα and IL1β genes are both conserved between zebrafish and mammals and most of the known components of their signaling pathways are also present in fish, including Myd88, caspase 1, and NFκB [190]. In zebrafish, two homologous copies of the TNF gene were identified, initially named *tnfα1* (TNFαa, referred to as TNFα in the rest of the text) and *tnfα2* (TNFαb) [191,192]. Orthologs of TNFRSF1B and TNFRSF1A have also been identified and their function has been studied in different processes [86,193]. The TNF pathway was shown to be crucial for host defense against mycobacteria [194,195,196,197]. In zebrafish embryos and larvae, IL1β is induced in response to injury [52,198] and to various infections [51,199,200,201,202]. In a model of chronic inflammation, knock-down of IL1β was shown to alter leukocyte recruitment to the site of inflammation [49]. Similarly, inhibition of the components of the IL1β pathway, such as caspase-1, reduces leukocyte recruitment to the wound and also NFκB activity [198]. IL1β also regulates infection-driven emergency myelopoiesis via NF-κB and C/ebpβ [203]. Finally, both TNFα and IL1β play important functions in regeneration [52,85,204]. Several transgenic zebrafish lines are available to study the roles of theses cytokines: a Tg(tnfα:GFP-F) line allows visualization of cells expressing TNFα due to the expression of farnesylated GFP driven by the tnfα promoter [50], and three distinct transgenic lines label cells expressing IL1β thanks to the expression of GFP under the control of the IL1β promoter [49,51,52]. A line expressing IL1β fused to mCherry in response to heat shock was also generated [52].

In mammals, another well-studied cytokine is Interleukin 6 (IL6), which harbors both pro-inflammatory and anti-inflammatory functions. IL6 binds to the IL6 receptor (IL6R) which triggers JAK/STAT-dependent downstream signaling cascades. IL6 and its receptors have been characterized in zebrafish [205]. The IL6 signal provides protection during Staphylococcus epidermidis infection [206], and it was also proposed to regulate regeneration processes [207]. Implication of the IL6 axis in regeneration is supported by the fact that STAT3, one of its downstream transcription factors, was shown to be required for the regeneration of adult heart [208] and of the larval optic tectum [209].

Orthologues of many other interleukins have been identified in zebrafish, including IL8, IL11, IL15, IL22, IL26, and IL34 [210]. Among the anti-inflammatory cytokines, IL4, IL10, and IL13 are probably the most studied ones. While IL10 limits the immune response to pathogens, preventing tissue damages, IL4 and IL13 stimulate type 2 immune responses. In zebrafish, IL10 and two IL4/13 loci (ohnologues), named IL4/13a and IL4/13b, were identified [211,212]. Thanks to mutant lines, IL4/13a and IL4/13b were shown to be required for the maintenance of T cells from the gills in a T helper 2 (Th2) cell-like phenotype and for the suppression of type 1 immune responses, while IL10 is essential for gill homeostasis and displays anti-inflammatory properties [106].

### 3.5. Chemokines

Chemokines constitute a subgroup of cytokines that display chemotactic properties toward leukocytes [213]. They are mainly produced in the thymus and in lymphoid tissues, from which they redistribute in the whole body on demand. They function both in homeostatic and pathological processes [214]. At resting state, they promote leukocyte homing in tissues where immune cells are required for development, function, and maintenance. In a disease context, they guide inflammatory leukocytes to the site of injury. They also regulate their differentiation, influencing both innate and adaptive immune responses. Chemokines are categorized in four subgroups, the two most populated ones being the CC and the CXC chemokines, while the C and CX_3_C families contain only a few members [213]. Their function is exerted through cognate cell-surface receptors from the G-protein Coupled Receptor (GPCR) family, classified with the same nomenclature as their chemokine ligands [213,214].

Chemokines and their receptors are found in all vertebrates as well as in some viruses [213,215]. Zebrafish chemokines have been searched since the 1990s [216]. A surprisingly large number of genes have been identified in this organism, with more than 80 putative genes for the CC subclass, highlighting numerous gene duplication events [217]. A novel lineage of the CXC family, restricted to teleost fish, has even been discovered, pointing out at a highly redundant chemokine system in zebrafish [217,218]. Although functional data are still lacking for most of these genes, quite a few chemokines and receptors displaying orthology with mammalian genes have been studied in more detail, including various isoforms of CXCL8, CXCL11, CXCL12, CXCL18, CCL2, and CCL25, and their associated receptors [216,219]. CXCL8a, the three CXCL8b variants, and CXCL18b activate the CXCR1 or CXCR2 receptors, abundantly expressed in neutrophils [81,82,89,91,220]. The CXCL11aa isoform transduces pro-migratory signals within macrophages through the classical CXCR3.2 receptor, while the atypical CXCR3.3 receptor may have antagonistic effects [221,222]. CXCL12a and b (also known as SDF1a and b) interact, respectively, with CXCR4b and a, the CXCL12a-CXCR4b couple having major involvement in inflammatory processes [92,223,224]. CCL2 specifically binds to the CCR2 receptor present on macrophages [225]. In addition, CXCR7, also termed ACKR3, seems to function as a scavenger receptor for both CXCL11 and CXCL12, as also evidenced in mammals [93,226].

Fluorescent reporter lines exist for CXCL18b, CXCL12a, and CXCR4b [82,83,84]. Fluorescent Timer (FT) transgenic lines expressing CXCR1 and CXCR2 fused to both GFP and tagRFP proteins have been generated to follow trafficking of these receptors in neutrophils, taking advantage of the different maturation time of the two fluorophores and of their differential stability in acidic environment [81]. A UAS-Gal4 transgenic line monitoring CXCR7 expression is also available [227]. Finally, morphants and/or CRISPR mutants providing global or cell-specific gene depletion have been described for CXCL8a/b, CXCL12a/b, CCL25a, CXCR1, CXCR2, CXCR3.2/3.3, and CXCR4a/b [89,90,91,92,93,94,228].

### 3.6. Interferons

Initially named for their antiviral activities [229], interferons (IFNs) are small secreted proteins (below 200 aa) specific to vertebrates [230]. Structurally, they belong to Class II helical cytokines [231,232]. IFNs have been classified in four types that were all present in primitive vertebrates, but while Type I and II IFNs are present in all groups of vertebrates, Type III is absent in teleosts and Type IV is absent in Metatheria and Eutheria mammals [230,233,234]. Types I, III, and IV IFNs are key players of innate immunity that are mainly produced by virally infected cells and, through induction of an inflammatory response, play an instrumental role in clearing viral infections. As a matter of fact, in case of an infection by a new virus, vertebrates cannot wait for the adaptive immune system to produce antibodies, and their survival mainly relies on virally induced IFNs [235]. Type II interferons have functionally diverged from other IFNs. They are rather immunomodulators, and, at least in humans, are instrumental to fight intracellular bacterial pathogens [236].

Type I interferons are the most diversified IFNs, with at least 17 genes in humans that all bind to the same receptor complex, IFNAR1-IFNAR2, but may have different biological activities due to specificities in their interactions with the two receptor chains [237]. In fish, we and others have described that type I IFNs may be classified in different groups that bind to at least two different receptor complexes [111]. Salmonids with at least 18 full-length genes have the highest number of Type I IFN genes [238]. Zebrafish contains four distinct Type I IFNs genes, one IFNAR1-related gene (CRFB5), and two genes equivalent to IFNAR2 (CRBF1 and CRFB2) [111]. In most vertebrate species, Type II INFs are encoded by a single gene called IFNγ, while teleost and some mammals have two genes encoding two different gamma IFNs. Type II interferon in mammals binds to the heterodimeric receptor IFNGR1-IFNGR2, while in teleosts, the two different Type II IFNs bind to different receptor complexes, containing CRFB6, CRFB13 and/or CRFB17 [239]. While mammalian species have up to four Type III IFNs genes, most vertebrates have a single Type III IFN gene that signals through its private receptor complex, IFNλR1-IL10R2, that shares one chain with type IV IFN receptor complex and one with that of other cytokines of the IL10 subgroup [233,240]. Teleosts do not have Type III IFNs. In zebrafish, the more recently discovered Type IV IFN (IFNυ) signals through the CRFB4-CRFB12 receptor that shares one chain with the Type III IFN receptor complex (CRFB4 equivalent to IL10R2) [233].

Structurally, the chains of IFN receptor complexes all belong to the class II helical cytokine receptor family and have a similar structure composed of an extracellular ligand binding domain, a transmembrane domain and a cytoplasmic domain involved in intracellular signaling mainly through the Jak/STAT pathway [241]. They induce the transcriptional activation of a vast set of Interferon Stimulated Genes (ISGs) that are mediators of inflammation and antiviral weapons. By comparing the zebrafish ISG repertoire to the human one, Levraud et al. have highlighted a core ancestral ISG set instrumental to fight viruses in vertebrates and a large lineage-specific ISG group, with no orthologs in other vertebrate clades, that probably represents lineage-specific strategies to fight viral infections [242]. Constitutive genetic activation of the interferon system is a major cause of Mendelian human auto-inflammatory diseases, also called interferonopathies [243].

Different zebrafish transgenic lines have been developed to study the interferon system either to label cells producing IFNφ1 [77] or IFNφ3 or to label cells stimulated by IFNs [78]. Interferons may easily be genetically over-expressed in zebrafish, and recombinant proteins can be produced and injected in zebrafish for functional studies [111,244]. Finally, various morphants and CRISPR mutants are available for both IFN and their receptor genes [78,86].

### 3.7. Inflammasomes

Inflammasomes are large complexes formed by multiple copies of three distinct proteins: a receptor, that recognizes intracellular danger signals (PAMPs or DAMPs); a pro-caspase, that gets auto-activated upon oligomerization once recruited within the active inflammasome; and an adaptor protein, that connects the activated receptor to the pro-caspase [245]. The primary aim of these cytosolic platforms is to activate the cysteine protease caspase-1 [246]. Activated caspase-1 will in turn cleave the pro-interleukins IL1β and IL18 into their active, pro-inflammatory forms. In addition, caspase-1 will trigger pyroptosis, a highly inflammatory form of programmed cell death, through proteolytic activation of pore-forming gasdermin.

Caspase-1-dependent inflammasomes are termed canonical [247]. Non-canonical complexes have also been described that result in the activation of other caspases, such as caspase-4, 5, 8, or 11. The precise mechanism of activation or the receptors involved in these non-canonical inflammasomes are not as well characterized as for the caspase-1-dependent complex. Nevertheless, they can also induce the release of various pro-inflammatory cytokines and DAMPs. Moreover, caspase-11 can trigger pyroptosis whereas caspase-8 is apoptotic.

In the canonical inflammasome pathway, the adaptor protein is always ASC (apoptosis-associated speck-like protein containing a caspase activation or CARD domain). On the other hand, the receptor belongs to one of the following families [245]: the NLR (nucleotide–oligomerization domain or NOD-like), the RLR (retinoic acid-inducible gene or RIG-like), or the ALR (AIM2-like) receptor family. NLRs contain leucine-rich repeat domains and can recognize a large range of danger signals, including LPS, bacterial lipopetides and toxins, or endogenous damage molecules. ALRs bind to DNA while RLRs detect viral RNAs and induce interferon-dependent pathways.

The inflammasome machinery is also present in teleost fish although individual components differ sometimes significantly from mammalian ones [248]. Four caspase-1-like genes are currently annotated in the zebrafish genome, only two of which have been linked to inflammasomes, caspa and caspb [249]. There is no clear orthology between these genes and mammalian caspase-1 since they both bear pyrin activation domains (PYDs) whereas mammalian caspase-1 possesses a CARD domain. CASPa has nevertheless been shown to be recruited within inflammasomes through the ASC adaptor, while CASPb can directly bind to LPS, suggesting that it may rather act as an ortholog of the non-canonical caspases [74,98]. A genuine ASC adaptor protein is present in zebrafish and is able to form ASC specks similarly to human ASC, albeit with subtle differences [74,250]. Another possible adaptor protein specific to zebrafish, CAIAP, has recently been described as important for inflammasome response during Salmonella Typhimurium infection [95]. The NLR family of receptors is not only present in zebrafish but it is also far more populated than the mammalian one, with more than 420 NLR genes currently identified in zebrafish [251]. Orthologs of NLRs such as NOD1, NOD2, NLRP1, or NLRC3 have been identified in zebrafish, but more than 400 of these NLR genes are unique to this vertebrate [100,252]. RIG-like receptors and their signaling pathways are on the other hand rather well conserved in zebrafish. Orthologs of RIG-I, MDA5, LGP2, MAVS, MITA, and TBK1 have been reported and a conserved role in interferon-dependent antiviral responses has been demonstrated [253,254,255,256]. Finally, two gasdermin genes, gsdmea and gsdmeb, were shown to promote cell death of microglial cells or pyroptosis of macrophages within Mycobacterium marinum-induced granulomas [75,99].

Various tools to study inflammasome biology in zebrafish have been developed, including morpholinos and CRISPR mutants for CASPa, CASPb, ASC, and the two gasdermins [248]. A CRISPR knock-in zebrafish line in which GFP has been fused to the endogenous ASC protein has also been generated, as well as a polyclonal antibody specific to zebrafish ASC, allowing to visualize ASC speck formation in real time [74]. Morpholino-based deletion mutants for NOD1 and NOD2 genes were successfully used to show that deficiencies in these NLRs increase the susceptibility to bacterial infections, as observed for patients suffering from IBD [100,101].

### 3.8. Proteases

Proteases produced by activated leukocytes and mast cells are also important contributors to inflammation, in particular neutrophil-derived serine proteases such as cathepsin G, neutrophil elastase and proteinase 3 [257]. These proteases are produced within azurophil granules, where they are present in their active form. In response to an inflammatory insult, and as a consequence of neutrophil activation, they become exposed on the extracellular neutrophil surface, following fusion of the exocytosed granules to the plasma membrane [258,259]. There, they retain their full biological activity and perform efficient proteolysis, allowing not only to kill bacteria in an infectious context [260,261], but also participating in various processes linked to sterile inflammation [262]. Indeed, serine proteases can cleave adhesion molecules, thereby facilitating neutrophil migration [263,264]. They can also control the activity of chemokines and cytokines, by proteolytic processing [265,266,267]. In addition, their binding to various cell surface receptors such as proteinase-activated receptors (PARs) or TLR4 activates downstream signaling pathways that lead to the upregulation and extracellular release of pro-inflammatory chemokines (e.g., CCL2, CXCL2, CXCL8) [268,269,270]. They have also been shown to cleave complement C5aR, thus inhibiting C5a-mediated pro-inflammatory signaling [271]. Finally, serine proteases have also been implicated in wound healing and induction of apoptosis [272,273].

Surprisingly, neutrophil serine proteases seem to be poorly conserved in fish and the loci in which they are normally found in mammalian genomes (i.e., the Chymase, Met-Ase, and Granzyme A/K loci) diverged substantially in fish species, including zebrafish [274]. Whether current phylogenetic analyses are incomplete or whether other serine proteases perform these functions in teleost fish remains to be determined.

### 3.9. Lipids

Over the course of the last decades, several studies have highlighted the role of poly-unsaturated fatty acids (PUFAs) and their derivates as bioactive lipids during inflammation. PUFAs are structural components of membrane phospholipids that are released upon tissue damage [275,276]. But they are also the substrates for specific enzymes producing bioactive lipids called lipid mediators that are structurally and chemically diverse [277]. Lipid mediators include the well-known eicosanoids (prostaglandins and leukotrienes) that are derived from arachidonic acid (AA) by the action of cyclooxygenase (COX) isoenzymes during the first phase of inflammation and which promote inflammation and regulate pro-inflammatory cytokine production [278,279]. In contrast, specialized pro-resolving mediators (SPMs) are enzymatically derived from AA, docosahexaenoic acid (DHA), or eicosapentaenoic acid (EPA), through the action of COX and lipoxygenases (LOX). They include lipoxins, families of resolvins, protectins, and maresins [277]. SPMs possess bioactions when administered in animal models and share the following properties: limitation of PMN infiltration, regulation of pro-inflammatory mediators, modulation of macrophage response, and decrease of disease severity. Therefore, they are potent regulators of the resolution of inflammation (for review [277,280]).

In zebrafish, the enzymes producing lipid mediators are present and were studied in the context of inflammation and infection. For COX/PGE2 pathway, the zebrafish possesses two copies of COX2: prostaglandin–endoperoxide synthase-2a and -2b (ptgs2a and ptgs2b genes). First, COX2 protein is induced in LPS-stimulated zebrafish embryos [281]. Second, using pharmacological inhibition of COX or CRISPR knockdown approaches, COX signaling was shown to be instrumental for the immune control of Aspergillus infection in the zebrafish larva, and exogenous prostaglandin E2 (PGE2) rescued the susceptibility to infection in this model [282].

LOX enzymes are encoded by the alox genes and alox5a, alox12, alox12b, alox15b, and alox2 are expressed in the zebrafish larva [283,284]. In the tail fin amputation model, Loynes and collaborators demonstrated that lipid mediators are crucial regulators of macrophages and neutrophils interplay at the wound [285]. Recently, protectin D1 (PD1), a potent modulator of the resolution phase of inflammation that displays protective properties and favors tissue regeneration in planarians [286,287], was shown to improve the regeneration of the caudal fin fold in the zebrafish larva by modulating macrophage function [288]. Lipidomic studies may improve in the future our knowledge concerning the lipid mediators produced during inflammation in zebrafish. SPMs have been proposed to signal through GPCRs: lipoxins, resolvins, maresins, and protectins bind to FPR2/ALX, GPR32, GPR18, chemerin1, BLT1, and GPR37 [289]. In zebrafish, only GPR18, chemerin1, BLT1, and GPR37 are present. Whether other GPCRs function as SPM receptors in zebrafish still needs to be investigated.

### 3.10. Other Pro-Resolving Mediators

While lipid mediators play a central role in the resolution of inflammation, several protein effectors are also actively involved in this process, including annexin A1 and galectins.

Annexin A1 is an important regulator in the glucocorticoid receptor pathways. Highly abundant in the cytosol of neutrophils and macrophages at resting state, it is secreted following cell activation and allows to revert the pro-inflammatory phase by (1) decreasing neutrophil infiltration in the wounded tissue; (2) promoting neutrophil apoptosis and clearance by macrophages; (3) downregulating pro-inflammatory genes in these immune cells; and (4) reprogramming macrophages toward a pro-resolving phenotype [290,291,292,293].

Galectins are lectins that are expressed by various immune cells and display specificity for β-galactose-containing oligosaccharides [294]. Fifteen distinct proteins are present in mammals and are categorized into proto, chimera, or tandem-repeat type, based on the number of sugar-binding domains they possess [295]. Although galectins participate in the initiation of inflammation [296], their most well-described role in inflammatory processes is to contribute to the resolution of inflammation by controlling neutrophil turnover, downregulating pro-inflammatory cytokines, enhancing the production of anti-inflammatory molecules, and facilitating macrophage switch toward anti-inflammatory state [294,297,298].

Eleven annexin genes have been reported in zebrafish [299]. The annexin A1 gene is present as four distinct isoforms (anxa1a, b, c, and d), all orthologous to human annexin A1. Several allelic mutants of these genes have been identified and the sa1750 mutant, which bears a premature stop codon in the anxa1c gene, presents a microglia with defective phagocytic functions [107]. Several galectin genes have also been characterized in this organism, including orthologs of galectins 1, 2, and 3 [300]. Morpholinos have been described for galectin1-like genes, allowing to study their role in cell differentiation and development [301,302].

## 4. Molecular Mechanisms and Signaling Pathways Governed by Inflammatory Mediators in Zebrafish

The increasing popularity encountered over the years by the zebrafish model has been a driving force to generate numerous models of inflammation. These models rely on various technical approaches: exposure to abrasive chemicals, ingestion, or injection of toxic compounds (small molecules or proteins), physical injuries (ablation, amputation, incision, laser injury, hot/cold shock), genetic mutations, and evidently, infections by various pathogens, including human-specific ones. They may mimic more or less closely human pathologies, but they all allowed to make valuable progress in the comprehension of the molecular mechanisms at play during the inflammatory response, both in zebrafish and, in a more global picture, in humans. Here, we highlight how these models have helped describe the role of the inflammatory mediators and precise their signaling pathways at various steps of the inflammatory process.

### 4.1. Induction of Inflammation

Among the early signals generated in response to a microbial or sterile injury, small molecules and free radicals are instrumental in enabling leukocyte recruitment and initiating the inflammatory response. In the last decades, zebrafish has helped gain valuable mechanistic insights on how and through which pathways these chemical signals trigger inflammation.

Nucleotides and calcium have been proposed to act as primary damage signals, while ROS would come as secondary messengers in the inflammatory process. Using the tail fin amputation model (Figure 2), Mulero and coworkers, for example, showed that both ATP and Ca^2+^-dependent signaling were able to induce H_2_O_2_ production and subsequent recruitment of leukocytes at the wound within 1.5 h post injury [112]. Using specific inhibitors of these different pathways, they elegantly demonstrated that ATP acts through P2Y purinergic receptors whereas intracellular calcium is raised through phospholipase C-dependent depletion of the ER stores. These two pathways synergistically contribute to the activation of the NADPH oxidase DUOX1, thereby producing H_2_O_2_ gradients that recruit neutrophils and induce NFκB signaling. In this setup, the ATP/P2Y signal seems to act prior to Ca^2+^ signal and may even directly contribute to the modulation of intracellular calcium levels [112]. Calcium would in turn act as the ultimate activator of DUOX1, through direct interaction with the EF-hand motifs of the oxidase. These different steps must occur very rapidly after injury since a noticeable gradient of H_2_O_2_ is already generated within 20 min by the DUOX1 enzyme present at the wound margin. Indeed, using calcium sensor, another study showed that wounding induced a very rapid and transient calcium wave that spreads within few minutes from the injury site. However, the interplay between Ca^2+^ signal and H_2_O_2_ is debated, as calcium inhibition via thapsigargin has no effect on H_2_O_2_ burst, suggesting that H_2_O_2_ and calcium signaling act independently [67].

In another study, Leite et al. demonstrated that the levels of extracellular ATP and ADP rapidly decrease through hydrolysis, following challenge with copper sulfate, while AMP, adenosine, and inosine levels slowly increase due to modulation of the activity of the AMP deaminase (ADA) [303], suggesting that ATP signaling occurs early in the inflammatory process since its degradation is also extremely rapid. In this injury model, they also observed a COX2-dependent increase of PGE2 shortly after Cu-stimulation, followed by a return to basal level and another increase one day after challenge. The first wave of PGE2 production is accompanied by neutrophil migration, activation of myeloperoxidase, and production of TNFα and IL1β, demonstrating that this lipid may also be involved in the induction phase of inflammation.

**Figure 2 biology-12-00153-f002:**
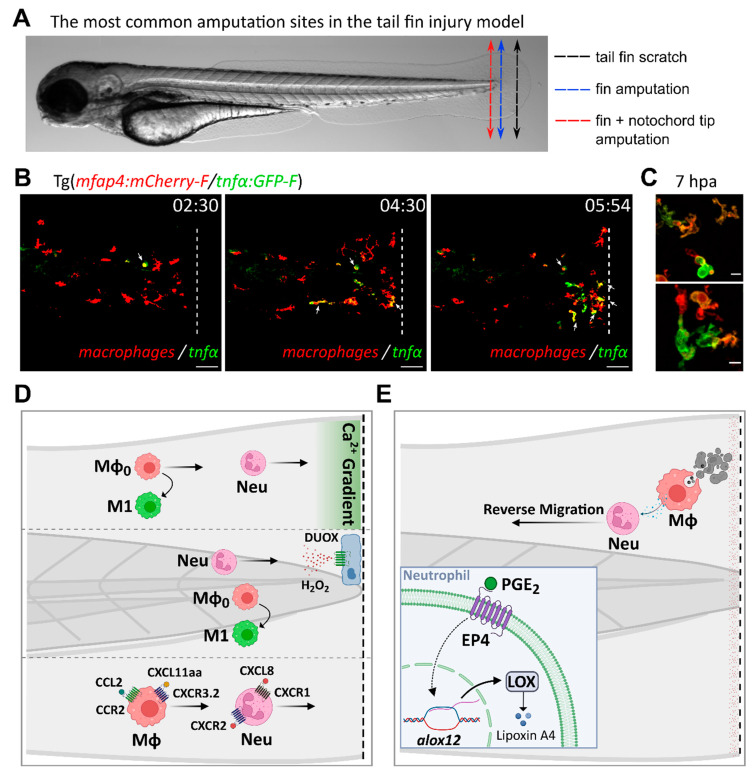
Inflammatory mediators and signaling pathways regulating leukocyte recruitment and activation during sterile injury, as inferred from the tail fin injury model. (**A**). The most common amputation/wounding sites used in the tail fin injury model are highlighted on a representative transmitted light image of a 3 dpf larva. Black dotted arrow: scratch on the tail fin tip. Blue dotted arrow: transection of the fin tip only, just before the notochord. Red dotted arrow: transection of the fin and the tip of the notochord. (**B**). Time-course of macrophage recruitment and activation following tail fin transection (intact notochord). Real-time imaging of the fin fold of a 3 dpf amputated larva from the Tg(*mfap4:mCherry-F*/*tnfα:GFP-F*) line, using confocal microscopy (unpublished data by Nguyen-Chi M. et al). In this line, macrophages and TNFα-expressing cells are labelled with red and green fluorescence, respectively. The representative frames show macrophage recruitment and activation toward an M1 phenotype at the wound, as previously observed by Sipka et al [53]. M1 macrophages, i.e., macrophages expressing TNFα, are indicated with white arrows. Time after amputation is indicated in hour:min. Dotted lines show the wound margin. Scale bar: 50 µm. (**C**). Representative images of activated macrophages at high magnification at 7 h post-amputation (unpublished data by Nguyen-Chi M. et al). Scale bar: 10 µm. (**D**). Tail wounding with fin amputation only. *Upper part:* fin wounding triggers instantaneous calcium flashes that are required for the recruitment of both neutrophils and macrophages. Ca^2+^ signaling is also important for macrophage polarization toward M1-like phenotype [53]. *Middle part:* reactive oxygen species H_2_O_2_ induces the recruitment of neutrophils at the wound site [112,113]. ROS are also required for macrophage M1-like activation [53]. *Lower part:* chemokines are essential for leukocyte recruitment to injury sites. Neutrophils migrate toward the wound via the CXCL8/CXCR1 axis while the CXCL8/CXCR2 axis promotes their dispersal [81,304]. By contrast, macrophage recruitment relies on CCL2/CCR2 and CXCL11aa/CXCR3.2 axes [221,225]. (**E**). Tail wounding with both fin and notochord tip amputation. Macrophage uptake of apoptotic cells triggers the resolution of inflammation by the production of PGE_2_ [285]. Mechanistically, PGE_2_ acts via EP4 receptors present on neutrophils, stimulating LOX expression. LOX activity then induces a lipid mediator switch that activates lipoxin production. Mϕ: macrophage. Neu: neutrophil.

More recently, Wittmann et al. also unravelled some of the mechanisms through which nitric oxide (NO) acts as an early pro-inflammatory signal in two distinct larval models of injury: a CuSO_4_-induced model affecting the peripheral sensory nervous system, and a model of epithelial injury based on incision of the ventral fin fold [305]. This study took advantage of an automated screen of FDA-approved drugs for repurposing in immunomodulatory applications and identified NO-dependent pathways whose inhibition led to decreased leukocyte infiltration and reduced inflammatory burden in the challenged tissues. Importantly, they found that NO synthase inhibition provides protective and anti-inflammatory effects in both wound settings. They also demonstrated that NO signaling results in increased protein S-nitrosylation. This later process may however be beneficial as Matrone et al. demonstrated that inhibition of the NO synthase iNOS results in reduced nuclear S-nitrosylation and impaired regeneration following tail fin amputation [306].

How NO production is triggered remains, however, unclear. Palmitate treatment of zebrafish embryos was reported to induce the synthesis of NO, as well as ROS and pro-inflammatory cytokines such as IL1β and TNFα, possibly in an NFκB-dependent manner [307]. The Battistini group developed a chromatographic method to measure NO levels in zebrafish larvae and showed that bacterial LPS and various small chemicals (sodium nitroprusside or copper ulfate) can also increase NO levels [308]. Interestingly, in a chemically-induced model of enterocolitis, Oehlers et al. observed that prolonged exposure to 2,4,6-trinitrobenzene sulfonic acid (TNBS) induces NO production around the notochord and the cleithrum, while larvae treated with dextran sulfate sodium (DSS) show no increase in NO levels, despite displaying otherwise similar inflammatory patterns in the intestine [309]. These subtle differences highlight the very fine tuning of the pathways regulating NO-dependent activation during inflammation.

Early inducers of inflammation also include protein effectors, in particular the complement anaphylatoxins C3a and C5a. Using LPS-challenged or tail fin-amputated larvae, Forn-Cuní et al. showed that the C3 paralogs C3.1, C3.2/3, and C3.6 are upregulated shortly after the injury, the peak in mRNA levels being reached 6 h post amputation (hpa), while continuous decrease is observed after this stage [102]. Increase in C3 components is associated with increased migration and infiltration of neutrophils, together with a strong upregulation of pro-inflammatory IL1β. Surprisingly, this study revealed that two other paralogs specific to teleost, C3.7 and C3.8, have rather opposite effects. Their upregulation first peaks up at 3 hpa, followed by a return to basal levels at 6 hpa and a second-wave upregulation 24 h after injury. This modulation of gene expression is accompanied by a reduction in neutrophil migration/recruitment and in IL1β production, suggesting that gene duplication events in zebrafish gave rise to teleost-specific C3 paralogs that have anti-inflammatory properties [102]. For C5, the Renshaw group showed that injection of recombinant zebrafish C5a anaphylatoxin in the otic vesicle enhances neutrophil recruitment and pro-inflammatory responses. They also developed an elegant model of humanized zebrafish expressing human C5aR and demonstrated that injection of human recombinant C5a in the otic vesicle has similar and even stronger effects, thereby highlighting how well-conserved the downstream signaling pathways of complement C5 are in zebrafish [23].

HMGB1 also plays an important role in triggering inflammation, especially in the brain where it promotes sustained neuroinflammation through the activation of microgial cells. Paudel et al. showed that inhibition of HMGB1 by the specific inhibitor glycyrrhizin (GL) has anticonvulsive effects and ameliorates the memory function of adult zebrafish in a model of epileptic seizure induced by pentylenetetrazole [310]. HMGB1’s deleterious effect in this model seems to be mostly dependent on its pro-inflammatory properties and is exerted through TLR4-dependent activation of NFκB, leading to an upregulation of pro-inflammatory cytokines such as TNFα (Figure 3A). Similar neuroprotective effects of GL are observed in a larval model of neuronal damage induced by the neurotoxin 1-methyl-4-phenyl-1,2,3,6-tetrahydropyridine (MPTP) [311]. Blockade of HMGB1 improves zebrafish locomotor function, length of dopaminergic neurons and counter-balances the loss of vasculature, all these MPTP-induced symptoms mimicking otherwise Parkinson’s disease-like conditions. In addition, the number of apoptotic cells and the levels of pro-inflammatory markers (IL1β, IL6, TLR4b, HMGB1a, NFκB) are reduced upon co-treatment with GL. Similarly, in an alcohol-induced model of skeletal muscle atrophy, the HMGB1/TLR4/NFκB axis was shown to be upregulated and led to IL1β and TNFα production, subsequently inducing ROS release and imbalance of the redox system through upregulation of antioxidant superoxide dismutases 1 and 2 (SOD1 and SOD2), as well as NOX2 [312].

Various actors are thus act concomitantly to initiate the inflammatory response, and interestingly, many models developed in zebrafish have highlighted the role played by the transcription factor NFκB in relaying these early inflammatory signals. NFκB-dependent transcriptional regulation of pro-inflammatory genes has indeed been evidenced in response to various stimuli and insults, including physical injuries such as tail fin amputation, as stated above, or exposure to microbial agents and toxic compounds. It is for example well-documented that TLR4 signaling following exposure to bacterial lipopolysaccharide (LPS) results in NFκB activation in zebrafish larvae [281]. Another example is given by the model of mucosal inflammation elicited by cigarette smoke extracts that cause marked inflammation in the gills, characterized by overexpression of TNFα, IL1β, and MMP9, and induce extensive morphological changes following prolonged exposure [313]. NFκB signaling has also been described in mutation-based models of inflammation. Mutations in the Atp1b1a subunit of the Na^+^/K^+^-ATPase generates a psoriasis-like phenotype reminiscent of skin malignancy due to appearance of epidermal aggregates (Figure 3B). This phenotype is dependent on the activation of the PI3K-AKT-mTORC1-NFκB-MMP9 pathway, which leads to the hyperproliferation of keratinocytes and subsequent invasion of the epidermal layers [314]. All together, these examples highlight how NFκB-dependent signaling acts as a central hub in inflammation (Figure 3).

**Figure 3 biology-12-00153-f003:**
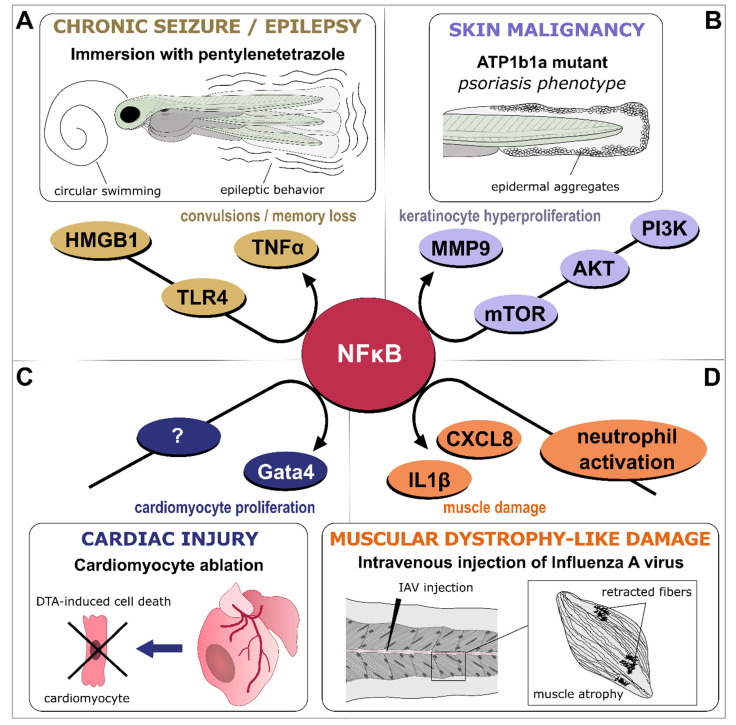
NFκB-dependent signaling as a central hub in inflammation. Various models of inflammatory-driven pathologies generated in larvae or adult zebrafish have identified NFκB signaling as part of the disease onset, progression, or recovery process. (**A**)**.** In an adult model of epileptic seizure induced by pentylenetetrazole, the HMGB1/TLR4/NFκB pathway generates deleterious inflammation that results in a convulsive behavior [310]. (**B**)**.** In larvae, *atp1b1a* mutation results in psoriasis-like phenotype and skin malignancy, through activation of the PI3K-AKT-mTORC1-NFκB-MMP9 pathway, which promotes keratinocytes overgrowth and invasiveness of epidermal layers [314]. (**C**)**.** In contrast to these deleterious effects, NFκB pro-inflammatory signaling is required for cardiac regeneration as exemplified in an adult model of genetic cardiomyocyte ablation where NFκB-dependent upregulation of *gata4*-controled genes permits cardiomyocyte proliferation and epicardial infiltration [315]. (**D**)**.** Infection of zebrafish larvae by Influenza A virus (IAV) induces a neutrophil-dependent activation of NFκB in skeletal muscles, upregulating pro-inflammatory cytokines such as IL1β and CXCL8 and generating sustained muscle damage that mimics Duchenne muscular dystrophy [316].

### 4.2. Neutrophil Migration and Activation

What mainly characterizes neutrophils is their capacity to move rapidly to sites of infection in tissues in order to deploy their antimicrobial arsenal. Neutrophil recruitment can be triggered by various chemotactic signals, including danger signals (ATP, H_2_O_2_, NO, or HMGB1), chemokines and cytokines (IL8, IL1β, or TNFα). Tracking neutrophils and delineating their migratory profile in the zebrafish larva have improved our knowledge on neutrophil biology during inflammation.

Caudal fin fold transection of the zebrafish larva triggers inflammation during which two most dominant types of innate immune cells—neutrophils and macrophages—are recruited to defend the wound against possibly invading pathogens and ensure proper tissue repair. After tail wounding, neutrophils are the first recruited and arrive at the injury site within 20 min [37]. Niethammer et al. showed that a gradient of H_2_O_2_ emanates at the wound margin and guides neutrophils within tissues via chemotaxis [113]. Mechanistically, H_2_O_2_ drives neutrophil attraction to the wound by inducing oxidation of the cytoplasmic factor Lyn, an Src family kinase (SFK) that acts as a redox-sensor in neutrophils [114]. A similar recruitment mechanism is evidenced in zebrafish larvae following genetic inhibition of TNFR2 or TNFα [193]. In these models, a strong inflammatory phenotype is observed in the skin, characterized by high levels of IL1β, TNFα, and prostaglandin-endoperoxide synthase 2b (PTGS2b or COX2b), as well as massive infiltration of neutrophils. The inflammatory response emanates both from infiltrated neutrophils and from skin keratinocytes, the latter being able to activate DUOX1, leading to an H_2_O_2_ gradient that is then sensed by neutrophils through Lyn kinase and induces TNFα production.

NO is another important driver of neutrophil migration. Using morpholino-dependent knock-down approaches in models of neuromast and epithelial damage, Wittmann et al. established that the constitutively expressed nos2b gene, coding for an ortholog of mammalian nitric oxide synthase 3 (NOS3, also known as endothelial NOS or eNOS), is in part responsible for the NO-dependent recruitment of leukocyte at the wound [305]. Inhibition of the NO sensor soluble guanylate cyclase (sGC) has similar protective action, suggesting a role of the downstream cGMP substrate in further propagating the inflammatory signal.

Besides TNFα, many cytokines and chemokines are also involved in the recruitment of neutrophils in zebrafish, such as IL1β and IL8 [49,198]. The case of the cytokine IL8 has been well studied and highlights the conservation of key pathways between mammals and zebrafish. Similarly to mammals, IL8 (or CXCL8) was shown to act as a potent chemoattractant for neutrophils in zebrafish. IL8 was shown to be upregulated at the infection site and at the wound site [89,90]. In addition, it is sufficient to instruct neutrophil migration in vivo [90]. The CXCL8/CXCR1 axis is required for neutrophil recruitment to wounds, while the CXCL8/CXCR2 axis is involved in neutrophil dispersal [81,304].

Recently, intravital imaging has highlighted a particular migratory pattern of activated neutrophils: a warm-like migration behavior, referred to as “neutrophil swarming” [317]. Studies using the zebrafish embryo showed that neutrophil swarming is a conserved process. Analysis of the early stages of neutrophil swarming in this model showed that this process begins with a pioneer neutrophil around which the swarms develop. The pioneer neutrophil releases NETs whose components are required for swarms [223]. Furthermore, the key to this process of swarming is the Ca^2+^/ATP signaling which controls the collective behavior of neutrophils at the wound. An ATP-dependent calcium alarm signal triggers neutrophil clustering at the site of damage. This “calcium alarm” signal spreads rapidly through the neutrophil cluster, thanks to connexin-43, which promotes the release of ATP, thereby increasing neutrophil chemoattraction [318].

### 4.3. Macrophage Recruitment and Polarization

Functionally plastic, macrophages can display many different roles. They have the ability to detect various signals present in the microenvironment and to respond to them by adopting equally diverse phenotypes, by a process termed polarization. For simplicity, macrophage polarization states have been classified into two categories: the classically activated M1 macrophages, which express proinflammatory cytokines and display a bactericidal activity toward pathogens, and the alternatively activated M2 macrophages, which have an anti-inflammatory function and are involved in healing and remodeling. Typically, M1 phenotype has been associated with the first phases of inflammation while M2 program is involved in the resolution phase [319]. However, this binary classification that recalls the pro and anti-inflammatory functions of Th1/Th2 is an oversimplification and does not reflect the wide diversity of macrophage phenotypes that can be found in vivo in complex environments [320].

Relative to the way of stimulation, polarized macrophages are characterized by the differential expression of cytokines, chemokines, receptors, and enzymes. The expression patterns of these macrophage-specific markers induce a myriad of functional profiles driving M1 or M2 responses. Some of these molecular markers are conserved between fish species, such as nitric oxide synthase 2 (NOS2), pro-inflammatory cytokines IFNγ, TNFα and IL1β, MHC class I and class II molecules, homologues of IL13 and IL4, arginase enzyme ARG2, and IL10 [321,322,323]. Furthermore, the complete transcriptome of larval zebrafish macrophages has been determined and gene expression profiling revealed high similarities in terms of gene expression with human macrophages, including M1 and M2 markers [324].

The transparent zebrafish embryo provides a unique opportunity to visualize macrophage polarization in real-time. Phenotype switching in individual macrophages has been observed by intravital imaging in the zebrafish larvae expressing reporters for macrophages and TNFα. Indeed, a few hours after tail wounding, macrophages expressing pro-inflammatory cytokines such as TNFα, TNFαb, IL6, and IL1β are recruited to the wound, resembling human M1 macrophages. Fate tracing of TNFα-positive macrophages revealed that these pro-inflammatory macrophages convert into a mixed phenotype expressing both M1- and M2 markers, highlighting that in vivo the process of macrophage polarization is highly dynamic [50].

Macrophage polarization results from a complex network of molecular mechanisms that have been intensively studied in vitro. However, the molecular determinants of polarization in vivo are still elusive. The zebrafish embryo fills this gap of knowledge, by making it possible to fully investigate macrophage activation dynamics in a whole organism in vivo, where tissues remain intact. In a recent study, upon fin fold amputation, macrophage activation toward M1 phenotype was shown to be mediated by early wound signals, calcium, and ROS, these two signals acting independently. While calcium triggers recruitment and M1 activation of macrophages, ROS promote M1 polarization though NFκB and Lyn [53]. Another study highlighted a new, hypoxia-dependent mechanism of M1 activation. In response to tail fin injury, the stabilization of the transcription factor hypoxia inducible factor (HIF) 1α activates COX2, leading to the production of PGE2. This HIF/COX/PGE2 pathway, which differs from better known DAMP/PAMP-mediated pathways, results in the upregulation of the pro-inflammatory cytokines TNFα and IL1β and activation of macrophages [325]. Macrophages adopt their function depending on the presence of inflammatory molecules in their microenvironment, and whose nature reflects the type of threat imposed to the damaged tissue. Interestingly, one study characterized different injury models in larval zebrafish and found that each damage model—thermal injury, infection or transection—triggers distinct inflammatory and tissue responses, highlighting the fine-tuned regulation of wound healing in vivo, highly context-dependent [326].

### 4.4. Resolution of Inflammation

Neutrophil removal at the wound is one of the key steps of the resolution phase. In mammals, it is commonly assumed that neutrophils at the wound undergo apoptosis and are taken up by macrophages initiating the resolution process [327]. In line with this idea, inhibition of caspase activity in tail fin-amputated zebrafish larvae increases the number of neutrophils remaining at the wound, suggesting that caspase-dependent apoptosis of neutrophils contribute to their removal during the resolution phase [39]. This view has, however, been challenged by the discovery that neutrophils migrate away from a wound and go back into the vasculature by a process called reverse migration [328]. A study using intravital microscopy to track naïve and reverse-migrated neutrophils revealed that reverse-migrated neutrophils exhibit an activated morphology keeping their capacity to respond to secondary insults or to bacterial infections [329]. At the molecular level, lipids and chemokines regulate neutrophil removal from the wound. Using knockdown approaches or exogenous PGE2, Loynes and collaborators demonstrated that macrophages produce PGE2, which drives neutrophil removal at the wound by promoting reverse migration. In this model of tail fin amputation, PGE2 signals through EP4 receptors leading to an increase in Lipoxygenase 12 (ALOX12) production and promoting resolution of inflammation. This interesting study proposed that Lipoxygenase 12 produces Lipoxin A4 (LXA4) that in turn regulates neutrophil migration [285]. Hypoxia and chemokine induction also play critical roles in the neutrophilic resolution. Constitutive activation of HIF1α decreases neutrophil apoptosis and reverse migration, resulting in a delay of inflammation resolution [57]. Similarly, the CXCL12/CXCR4 axis participates to the retention of neutrophils at the wound and impairing this signaling pathway promotes neutrophil dispersal [223].

While pro-inflammatory macrophages have been described in various systems in zebrafish, anti-inflammatory macrophages remain poorly characterized. This may be due to their versatile nature and the poor conservation of their markers between species. Reporter lines to visualize macrophage M2 states in zebrafish are still missing. However, one recent study showed a shift of macrophage populations from the inflammatory to the regenerative phase following cardiac cryoinjury, and they identified a macrophage subpopulation expressing Wilms Tumor 1b (WT1b) that accumulates within regenerating tissues, suggesting a role for WT1b in organ regeneration [330]. This macrophage population displays a unique signature of genes involved in tissue homeostasis restoration and extracellular matrix remodeling and might represent an M2-like population of macrophages.

Due to the complex and specific cellular networks that mediate the resolution of inflammation, single-cell RNAseq (scRNAseq) analyses are perfectly suited to explore macrophage heterogeneity and determine signature of unique macrophage populations at play in specific inflammatory context. For instance, taking advantage of the short time frame of the regenerative process following sensory hair cell ablation, Denans et al. demonstrated that a single population of macrophages is recruited within neuromasts and undergoes three consecutive waves of transcriptional reprogramming as part of their anti-inflammatory activation sequence [331]. First, the Glucocorticoid (GR) pathway is activated, concomitantly to the upregulation of macrophage anti-inflammatory markers and genes involved in apoptotic control or negative regulation of ROS. Within 1 h after injury, IL10-signaling pathways become next activated. Finally, 3 h post wounding, upregulation of IL4- and polyamine signaling pathways allows inhibition of pro-inflammatory cytokines and induces oxidative phosphorylation within macrophages, indicating a switch toward repair functions [331]. Further investigations aimed at characterizing wound- or trauma-associated macrophages will be needed in the future to better understand macrophage heterogeneity and the role of different subsets during wound healing and regeneration.

### 4.5. Regeneration

While all animals have the capacity to heal wounds, only few species can fully regenerate their organs or lost body parts after an insult. This is the case for zebrafish, which have the remarkable capacity to regenerate numerous organs such as the heart, brain, spinal cord, retina, and fins, restoring the mass and function of damaged tissues. This regeneration process relies on migration and proliferation of differentiated cells or on proliferation of dedifferentiated cells which, after re-differentiation, are capable of giving rise to the new tissues. Recently, numerous studies highlighted the crucial role of the immune system and inflammation in this process, and few reviews have summarized the significant discoveries and advances made in the field [332,333].

Using the model of caudal fin fold regeneration in the zebrafish larva, several studies have questioned the role of macrophages in the regeneration process. Suppression of macrophages in larvae, using either the Metronidazole/Nitroreductase system or liposome-encapsulated clodronate treatment at different time points of the regeneration process, results in the impairment of fin fold regeneration [85]. Similarly, in the cloche mutants, which lack most hematopoietic cells, regeneration is affected and is accompanied by apoptosis [334]. Macrophages are known sources of important pro- and anti-inflammatory factors. In the case of cloche mutants, it has been proposed that myeloid cell depletion results in a prolonged expression of IL1β in the epidermal cells of the wound. Morpholinos knocking down IL1β expression or dexamethasone, a synthetic glucocorticoid that suppresses inflammation, lead to a reduction in apoptosis of regenerative cells, suggesting that a transient inflammation mediated by IL1β is required for cell survival at the wound [52]. In addition, early wound-macrophages are the major source of TNFα in amputated larvae. Blocking TNFα signaling pathway using pharmacological inhibition or morpholinos targeting either the receptor TNFR1 or the ligand TNFα, also affects fin fold regeneration, suggesting that macrophage-derived TNFα is a necessary signal that creates a permissive environment for regeneration [85]. On the other hand, persistent presence of TNFα-positive macrophages at the wound was reported to negatively correlate with regeneration [326]. Altogether these studies highlight the importance of a tightly regulated inflammation response in the regeneration.

The mechanisms driving regeneration of caudal fin in zebrafish larvae are reminiscent of what happens during spinal cord regeneration where peripheral macrophages control axonal regeneration by producing pro-regenerative TNFα and by reducing levels of IL1β [204]. In this model of spinal cord injury, Cavone and collaborators identified a specific population of pro-regenerative macrophages that interact directly with spinal progenitor cells upon injury [204]. This population of macrophages has a unique activation state characterized by the expression of TNFα and is localized in the vicinity of ependymo-radial glia (ERG) progenitor cells. Blocking TNFα pathway using pharmacological inhibition of TNFα release, CRISPR-mediated TNFα gene editing, or a stable TNFα mutant, results in the impairment of neurogenesis. TNFα was proposed to signal through TNFR1 on ERG progenitors promoting expression of regeneration-associated genes and stimulating neurogenesis [22].

Another important player in neuronal regeneration is HMGB1, as it was shown to enhance neurovascular remodeling, thereby permitting recovery of the locomotor function. In adult zebrafish, Fang and colleagues showed that HMGB1 contributes to regeneration after spinal cord transection, thanks to a tightly controlled modulation of the expression and localization of the protein following injury [335]. Shortly after transection (within 1 day), HMGB1 is downregulated, to avoid deleterious inflammatory signaling, and translocates in the meantime from the nucleus to the cytoplasm of motor neurons. In the time course of recovery, HMGB1 levels increase again both in the cytoplasm and nucleus, expression in the cytosol being completely abolished after 21 days. This modulation permits regrowth of axons and improvement of the swimming capabilities. In addition, HMGB1 is upregulated in endothelial cells, increasing angiogenesis and permitting proper revascularization of the neuronal tissues.

Activation of key inflammatory mediators/pathways is also a prerequisite for effective zebrafish heart regeneration. Transcriptomics analyses in adult fish indeed revealed that many genes classified under the ontology term “Inflammatory response” are upregulated following ventricular resection [336]. These include complement receptor C5aR1 whose downstream pro-inflammatory signaling is important for cardiac regeneration since its inhibition by the antagonist PMX205 yields a reduced number of proliferating cardiomyocytes [336]. In a model of genetic ablation of cardiomyocytes triggered by diphteria toxin A (DTA), Poss and coworkers demonstrated that activation of NFκB within cardiac muscle cells lining the injury site is required for proper regeneration [315]. Inhibition of NFκB signaling by conditional expression of the lκB superrepressor on the other hand leads to impaired regeneration and scar tissue, due to improper modulation of gata4-dependent genes, resulting in decreased epicardial cell infiltration at the wound and reduced cardiomyocyte proliferation (Figure 3C). The Janus kinase 1/signal transducer and activator of transcription 3 (JAK1/STAT3) pathway also conditions cardiomyocyte proliferation after injury, through upregulation of cytokines like IL11a/b or leukemia inhibitory factor LIF [208]. Altogether, these studies strengthen the concept that inflammation is a central and indispensable phase in the regenerative process, providing that it is tightly and timely controlled.

### 4.6. Infection-Driven Inflammation

The transparent zebrafish embryo model has become popular to study inflammation driven by host-pathogen interactions, thanks to feasible visualization of host phagocytes and pathogens in real-time using non-invasive intra-vital imaging. Over the years, many different models of infection have been generated in this vertebrate, not only for pathogenic bacteria, but also for viral and parasitic microbes, regardless of whether they are natural pathogens of zebrafish or mammal-specific pathogens. Routes of infection in these models are also quite diverse, ranging from natural mode of administration through immersion of the pathogen in the fish water to forced administration by oral gavage, microinjection in a localized tissue/organ, or microinjection directly into the blood circulation (Figure 4A). The latter strategies are more commonly used for non-natural fish pathogens, for which natural entry routes are not present on zebrafish epithelia.

Studying how the innate immune cells respond to pathogens helped gain insights into inflammatory signaling pathways and the role of pro-inflammatory cytokines. For example, a zebrafish model of Burkholderia cenocepacia infection revealed that B. cenocepacia requires macrophages, but not neutrophils, for efficient replication and induction of a fatal IL1β-dependent inflammatory response [337]. The pro-inflammatory cytokine IL1β has also been shown to be crucial to induce neutrophil-dependent chronic inflammation in a notochord infection model in zebrafish [49]. Moreover, the TNFα/IL8 inflammatory axis is instrumental for protective immunity against Mycobacterium abscessus in zebrafish larvae, by promoting neutrophil-dependent formation of granulomas [194]. Septins, which are major components of the cytoskeleton, are also important players in the immune defense against Shigella in zebrafish and they have been shown to be essential to restrict IL1β-mediated inflammation and sustain neutrophil defense in this model [200].

**Figure 4 biology-12-00153-f004:**
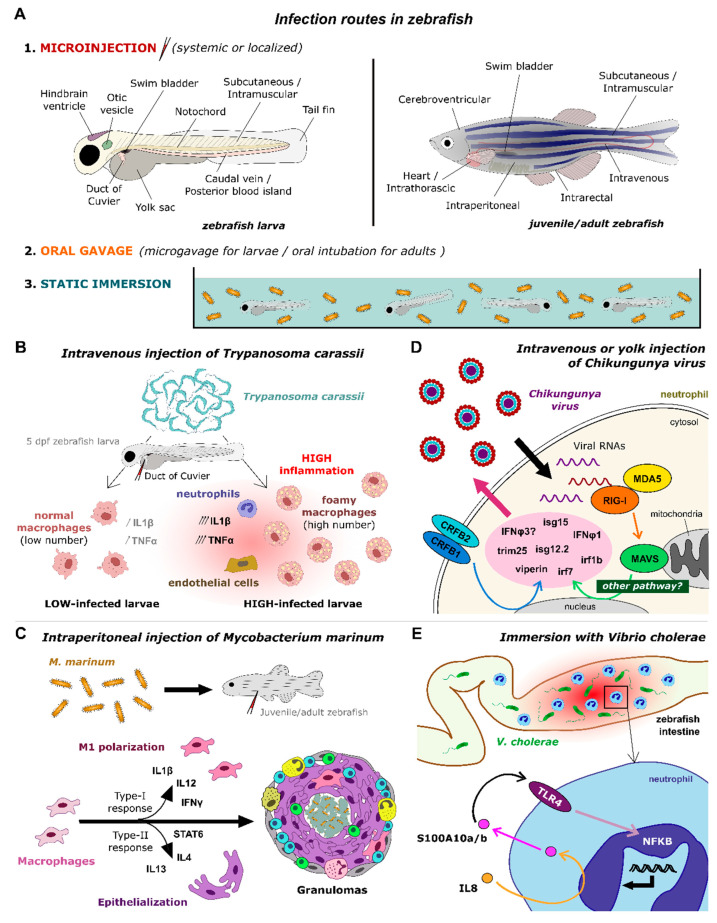
Signaling pathways unraveled by infection models in zebrafish. (**A**). Infection routes used in zebrafish models include micro-injection in various organs/tissues, oral gavage, and immersion of the pathogens in the fish water, both for larvae and adult zebrafish. (**B**). A model of infection by *Trypanosoma carassii* through intravenous injection in 5 dpf larvae revealed two distinct macrophage pro-inflammatory phenotypes depending on the level of infection, high-infected larvae producing foamy macrophages with strong pro-inflammatory response and incapacity to control the parasitemia [199]. (**C**)**.** A model of infection by *Mycobacterium marinum* through intraperitoneal injection in juvenile or adult zebrafish showed that both M1- and M2-type macrophage responses are needed for mycobacterial pathogenesis and formation of granulomas [338]. (**D**). A model of infection by Chikungunya virus through intravenous or yolk injection in larvae uncovered the interferon-dependent antiviral immune response elicited by neutrophils [77]. (**E**). An adult model of infection by *Vibrio cholerae* through immersion evidenced the NFκB-dependent recruitment of leukocytes in the intestine through activation of an IL8/S100/TLR4 pathway [339].

Recently, Jacobs et al. established a zebrafish model of infection with trypanosome and investigated the early innate responses of zebrafish to trypanosome infection [199]. Using a clinical scoring system of infected larvae to monitor parasitaemia development, the authors studied immune responses in high- or low-infected individuals. Independent to the number of trypanosomes, Trypanosoma carassii infection leads to a strong macrophage response, characterized by macrophage recruitment and division, and myelopoiesis. Interestingly, this study also revealed the occurrence of pro-inflammatory foamy macrophages in high-infected larvae (Figure 4B). Indeed, in high-infected larvae, uncontrolled parasitaemia induces a strong pro-inflammatory response, defined by an overall induction of TNFα and IL1β in both immune and endothelial cells, associated with susceptibility to the infection. In contrast, low-infected larvae show lower TNFα and IL1β responses and were able to control parasitaemia and recover from the infection [199]. Further studies are needed to explore how foamy macrophages are formed in vivo and which trypanosome components trigger M1 activation of these macrophages.

While M1 program is classically associated with the control of bacterial infections, the zebrafish Mycobacterium marinum model highlighted in a recent study the role of M2 phenotype in mycobacterial pathogenesis [338]. Indeed, although mycobacterial granulomas were thought for a long-time to be driven by type 1 inflammatory process, single-cell RNAseq of zebrafish granulomas revealed that both type 1 (IFNγ, IL12, IL1β) and type 2 responses (IL4, IL13) are induced (Figure 4C). Within zebrafish granulomas, Cronan et al. observed that macrophages expressing pro-inflammatory markers are not epithelialized whereas epithelialized macrophages express M2-associated markers, revealing an inverse relationship between M1-macrophage polarization and epithelialization. Moreover, they showed by deleting STAT6, a type 2 inflammatory transcription factor acting downstream of IL4, that formation of organized mycobacterial granulomas and macrophage epithelialization requires STAT6-dependent type 2 immune signaling [338].

Modeling of viral infections has also been successfully achieved in zebrafish over the past years and has contributed to unravel the inflammatory pathways activated during the antiviral response, notably the IFN- and inflammasome-dependent pathways. For example, Palha et al. showed that infection of zebrafish larvae by the Chikungunya virus induces a strong type I interferon response that is essential for fish survival, as demonstrated by the high lethality of fish where the two IFN-I-specific receptor chains, CRFB1 and CRFB2, had been knocked-down [77]. This IFN-dependent response is mainly elicited by neutrophils and leads to the upregulation of IFNφ1, IFNφ3, and various interferon-stimulated genes (ISGs) such as viperin, ISG12.2, ISG15, or TRIM25, through CRFB1/2- and possibly MAVS-dependent pathways (Figure 4D), although involvement of RIG-I and the downstream MAVS was not confirmed in a follow-up study relying on microarray gene expression analysis [340]. Intravenous injection of the spring viremia of carp virus (SVCV) into the caudal vein of zebrafish larvae also leads to an IFN-dependent antiviral response supported by CRFB1 and CRFB5 [341], while an immersion-based model of SVCV infection fails to activate interferon-dependent signaling pathways but still elicits a strong pro-inflammatory response by inducing IL1β, TNFα, and lymphotoxin α (LTα) [342]. Interestingly, high production of TNFα in this SVCV immersion model seems to be beneficial to the virus as it impairs host autophagy and viral clearance [343]. These contrasting results highlight the influence of the virus entry mode on the pathways to be activated.

Other virus models have also been efficiently used to dissect the TLR- and RLR-dependent pathways in zebrafish. For instance, zebrafish larva infection by snakehead rhabdovirus leads to the upregulation of TLR3 and TRAF6 [344] and demonstrates the protective role of the MDA5 receptor through induction of IFNφ1 [256]. Adult zebrafish challenged with viral hemorrhagic septicemia virus (VHSV, intraperitoneal injection) show increased levels of IFNφ1, IFNγ1, TNFα, and myxovirus resistance protein 1 (MXa) in the kidney, through TLR3-dependent activation [345]. In another study, Gong et al. showed that the RIG-I-like receptor LGP2 is critical for zebrafish survival upon infection by SVCV, as it promotes a strong IFN response through upregulation of IFNφ1, IFNφ3, MXb, IRF3, and IRF7, at early stages of the infection, and then downregulates this same response at later stages, by attenuating TBK1 and IKKi, upon sensing of high levels of IFN production [346]. This study thus highlights how LGP2, through this switch mechanism, can both act as positive and negative regulator of the antiviral response, firmly demonstrating that a single molecular actor can induce both pro-inflammatory and anti-inflammatory effects.

Finally, some of the models of viral infections developed in zebrafish generate an inflammatory burden and a damage to tissues that mimic certain human pathologies. A remarkable example is given by the model of infection by Influenza A virus (IAV) developed by Goody and coworkers, in which intravenously-injected IAV diffuses to and infects muscle cells, causing muscle fiber retraction [316], as seen in models of congenital muscle disease such as the Duchenne muscular dystrophy. IAV infection is associated with a strong overall antiviral response, leading to upregulation of IFNφ1 and MXa, and a local pro-inflammatory response in muscles, characterized by the NFκB-dependent induction of IL1β and CXCL8 (Figure 3D). Furthermore, neutrophils are abundantly recruited in muscle tissues, and more particularly at the detached end of the damaged fibers, suggesting a role of these immune cells in muscle degeneration [316].

While the zebrafish signaling pathways linked to pro-inflammatory cytokines, chemokines, and interferons are more documented, less is known about the early actors of inflammation, including alarmins and alarmin-like antimicrobial peptides. Nevertheless, preliminary studies have shown, for at least some of these actors, that they may function similarly to their mammalian counterparts. In a recent study, Zhang et al., for example, showed that in specific immune tissues (gills, gut, spleen, and kidney), most S100 genes are upregulated in response to bacterial challenge with E. tarda, or viral challenge with poly(I:C) or SVCV [347]. The Withey group also evidenced an overexpression of S100A10a/A10b in the intestine, both in an adult model of infection by the human intestinal pathogen adherent invasive E. coli (AIEC) [348], and in a larval model of infection by various Vibrio cholerae strains [339]. Interestingly, in the Vibrio infection model, upregulation of the calprotectin-like S100A10a/A10b couple is dependent on IL8/CXCL8, leads to activation of the TLR4 and NFκB pathways, and is concomitant to an increased production and recruitment of leukocytes, mostly neutrophils [339] (Figure 4E).

Much progress still remains to be achieved to fully comprehend the mode of action of these different effectors. Nevertheless, as shown in these few examples, modeling microbial infections in zebrafish has already considerably deepened our understanding of the inflammatory responses induced by pathogens, both of bacterial, parasitic, or viral nature.

## 5. Concluding Remarks

Over the years, the zebrafish has emerged as a remarkable animal model to study innate immune responses and inflammatory processes. Its numerous advantages over many mammalian models, including its transparency and genetic adaptability, have enabled increasingly sophisticated mechanistic studies that have contributed to a better understanding of the molecular events that orchestrate the different phases of the inflammatory response and the key effectors that ensure proper transition from the acute phase of inflammation to resolution and tissue repair. Undoubtfully, these models generated to study inflammation in zebrafish offer valuable assets for pharmacological development, notably due to the high similarity of the molecular actors and signaling pathways involved as compared to those described in humans.

In the future, manipulating pro- and anti-inflammatory mediators with help of inhibitors or enhancing compounds may be the key to therapeutical interventions aiming at reducing the inflammatory burden and restoring homeostasis in various inflammatory conditions affecting humans. In that respect, the zebrafish model opens up new possibilities for rapid and high-throughput screening of novel anti-inflammatory drugs, as well as evaluation of their cytotoxicity, providing an appealing alternative to rodent models for pre-clinical studies which may considerably ease the first steps of the drug development process.

## Figures and Tables

**Figure 1 biology-12-00153-f001:**
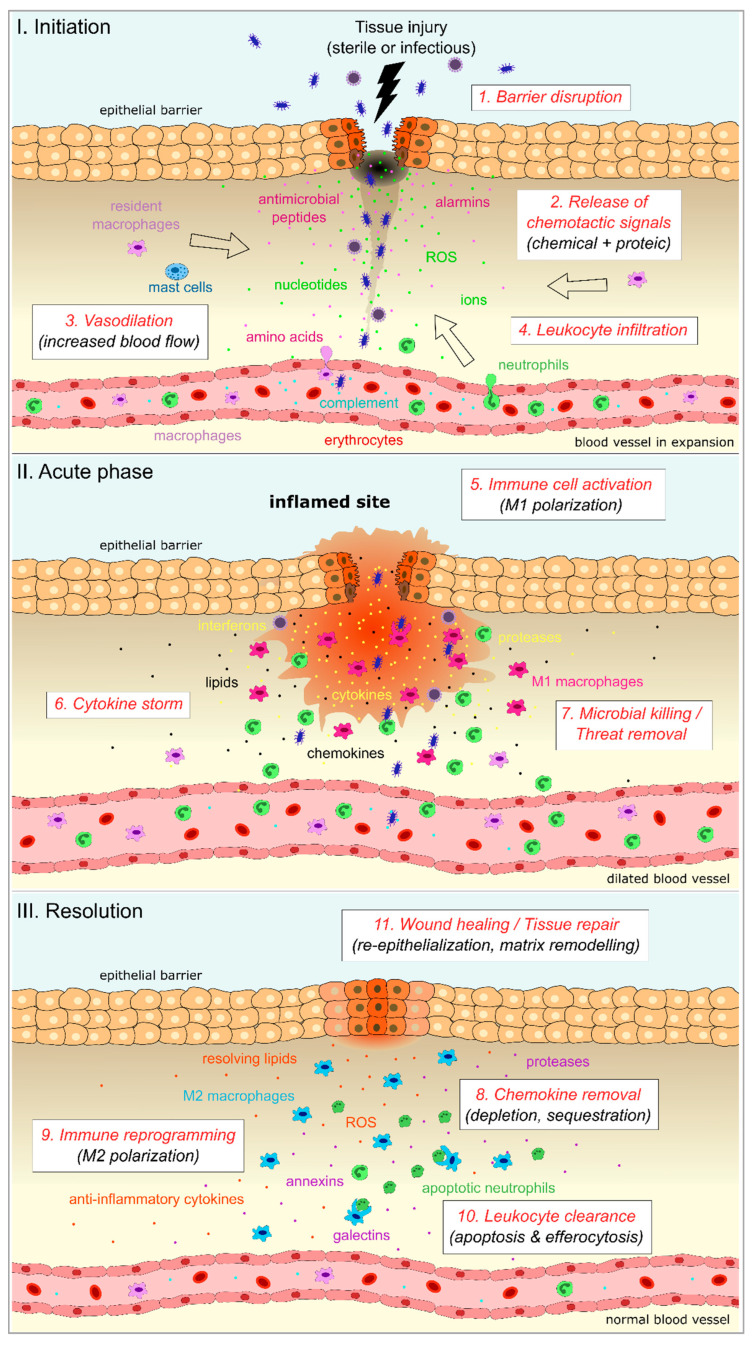
Cellular and molecular actors of inflammation during the three phases of the inflammatory response. I-Initiation: upon sterile or infectious tissue injury, disruption of the epithelial barrier leads to the release of early inflammatory signals that can be either chemicals or proteins. These signals modulate the blood flow by increasing vasodilation and generate chemotactic gradients allowing for immune cell infiltration within the injured tissue. Unfilled arrows indicate the direction of migration for resident and infiltrated immune cells. II-Acute phase: once recruited at the site of injury, immune cells get activated into a pro-inflammatory phenotype, leading to massive production of pro-inflammatory mediators (“Cytokine storm”) and threat removal. III-Resolution: tissue repair is then initiated by chemokine depletion (through proteolytic degradation or receptor-dependent sequestration) and immune cell reprogramming toward an anti-inflammatory/pro-resolving phenotype. This switch ends the storm, leads to leukocyte clearance and healing, and ultimately restores homeostasis.

**Table 2 biology-12-00153-t002:** Tools generated to study the molecular components of inflammation in zebrafish, including transgenic reporter lines, morpholinos, mutants, recombinant proteins and antibodies, overexpression systems, and fluorescent sensors. Tg: transgenic line; MO: morpholino.

Tool Name & Category	Target Component	Application/Target Type	References
Transgenic lines			
Tg(*6xNF-kBre:eGFP*)nc1	NFκB activation	Monitoring of the temporal and spatial patterns of NFκB activation	[73]
Tg(*asc:asc-eGFP*)	Inflammasome activation	Visualization of cells with inflammasome activated, CRISPR knock-in line in which the endogenous ASC is fused to eGFP	[74,75]
TgBAC(*mpx:GFP*)i114xTg(*lyz:h2az2a-mCherry,cmlc2:GFP*)	Neutrophil Extracellular Traps (NETs)	Visualization of neutrophils and neutrophil derived DNA-associated proteins	[76]
Tg(*ifnφ1:mCherry*)	Cells producing IFNφ1	Reporter line, *ifnφ1* promoter	[77]
Tg(*cryaa:DsRed/mxa:mCherry-F*)	Cells stimulated by IFNs	MXA promoter used to drive the specific expression of membrane-targeted mCherry in cells responding to type I IFN	[78]
Tg(*actb2:GCaMP3*)	Ca^2+^	Monitoring of Ca^2+^, using a line in which the GCaMP3.1 indicator is expressed under the control of ubiquitous β-actin promoter	[67,79]
Tg(*ins:hmgb1-eGFP*)	HMGB1	Dynamics of HMGB1 in β-cells	[80]
Tg(*lyz:cxcr1-FT*) FT: Fluorescent timer	CXCR1 in neutrophils	Constitutive and ligand-induced dynamics of CXCR1 in neutrophils	[81]
Tg(*lyz:cxcr2-FT*) FT: Fluorescent timer	CXCR2 in neutrophils	Constitutive and ligand-induced dynamics of CXCR2 in neutrophils	[81]
Tg(*cxcl18b:eGFP*)	CXCL18b	Reporter line, *cxcl18b* promoter	[82]
Tg(*sdf-1a:DsRed2*)	CXCL12a/SDF1a	Reporter line, *cxcl12a* promoter	[83]
Tg(*cxcr4b:mCherry*)	CXCR4b	Reporter line, *cxcr4b* promoter	[84]
Morpholinos (MO)			
MO-*tnfα*	TNFα	Cytokine	[85]
MO-*il1b*	IL1β	Cytokine	[49,52]
MO-*tnfrsf1b/tnfrsf1a*	TNFα receptors	Cytokine receptor	[86]
MO-*hmgb1a*	HMGB1a	Alarmin	[87,88]
MO-*cxcl8a*	CXCL8a	Chemokine	[89,90]
MO-*cxcr1*	CXCR1	Chemokine receptor	[91]
MO-*cxcr4b*	CXCR4b	Chemokine receptor	[92,93]
MO-*ccl25a*	CCL25a	Chemokine	[94]
MO-*asc*	ASC	Inflammasome	[74,75,95]
MO-*caspa*, MO-*caspb*	CASPa, CASPb	Inflammasome	[75,96,97,98]
MO-*gsdmea*, MO-*gsdmea*	Gasdermins	Inflammasome	[75,99]
MO-*nod1*, MO-*nod2*	NOD1, NOD2	Inflammasome	[100,101]
MO-*C3a*, MO-*C3aR*	C3a, C3aR	Complement	[102,103]
MO-*s100a10a*	S100A10a	Alarmin	[104]
Mutants and CRISPR Mutants			
*tlr2* mutant	TLR2	Receptor	[105]
*myd88* mutant	MYD88	Adaptor protein	[105]
*il4/13a or il4/13b mutant*	IL4/13a & IL4/13b	Interleukin	[106]
*il10*	IL10	Interleukin	[106]
*anxa1c* mutant	ANXA1C	Annexin	[107]
*ifnφ3^ip7/ip7^* mutant	IFNφ3	Interferon	[78]
*cxcr1* mutant	CXCR1	Chemokine receptor	[81]
*cxcr2* mutant	CXCR2	Chemokine receptor	[81]
*asc* mutants	ASC	Inflammasome	[99,108]
*caspa* mutant	CASPa	Inflammasome	[74,75]
*caspb* mutant	CASPb	Inflammasome	[109]
*s100i2 CRISPR*	S100I2	Alarmin	[110]
Recombinant proteins/antibodies			
Recombinant IFN*φ1*, IFN*φ2*, IFN*φ4*	IFNφ1, IFNφ2, IFNφ4	Production of recombinant IFN*φ* and injection in zebrafish for functional studies	[111]
Asc antibody	Asc	Inflammasome	[74]
Overexpression plasmids			
*IFNφ1-*pTol2S263C *IFNφ2-*pTol2S263C *IFNφ3-*pTol2S263C *IFNφ4-*pTol2S263C	IFNφ1 IFNφ2 IFNφ3 IFNφ4	Overexpression of the four IFNφ genes in zebrafish, using the pTol2S263C vector	[111]
Fluorescent sensors			
CellRox & acetyl-pentafluorobenzene sulphonyl fluorescein	ROS	ROS-specific probes that fluoresce in the presence of ROS	[53,112]
Fluo-3-AM	Ca^2+^	Ca-specific fluorescent probe for the detection of intracellular Ca^2+^ concentration	[53]
HyPer mRNA	H_2_O_2_	Genetically encoded ratiometric H_2_O_2_ sensor to monitor intracellular H_2_O_2_ concentration	[113]

## Data Availability

Not applicable.

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
