# Peer review of "Molecular Actors of Inflammation and Their Signaling Pathways: Mechanistic Insights from Zebrafish"

_biology, 2023, doi:10.3390/biology12020153_

Round 1

Reviewer 1 Report

In this manuscript, Leiba et al. summarized previous studies of immune response and inflammation in zebrafish, and presented a great resource review to the research community. This review article is well organized and comprehensively written, including components and mediators of inflammation, and pathways involved during inflammation. The authors further provided detailed lists of transgenic tools and mutants that have been developed using zebrafish, and such lists are very valuable for the scientific community to facilitate research designs using zebrafish as a model.

Overall, I believe this manuscript is a significant contribution to the field, and I strongly support it for publication.

Author Response

Response to reviewers’ comments on Manuscript Biology-2127022

We thank the reviewers for their appreciation of our work and for their very constructive comments that helped us improve our manuscript. We have taken into consideration all the concerns raised by the reviewers and modified the text and figures accordingly. Please find below a detailed, point-by-point response to the reviewer’s comments.

---------------------------------------------------------------------------------------

Response to Reviewer #1's comments

Point 1: in this manuscript, Leiba et al. summarized previous studies of immune response and inflammation in zebrafish, and presented a great resource review to the research community. This review article is well organized and comprehensively written, including components and mediators of inflammation, and pathways involved during inflammation. The authors further provided detailed lists of transgenic tools and mutants that have been developed using zebrafish, and such lists are very valuable for the scientific community to facilitate research designs using zebrafish as a model.

Overall, I believe this manuscript is a significant contribution to the field, and I strongly support it for publication.

Response 1: we thank the reviewer for his/her very positive comments on our manuscript and we hope this review will indeed be valuable for the research community working in the field.

Reviewer 2 Report

For readers who have a basic knowledge of inflammation and the advantage of using zebrafish as a research model, this article will attract the best attention if its major content begins from lines 243, as the transparency of zebrafish at embryonic stages and molecular techniques available can reveal very early events in inflammation triggered either by infection or tissue damage. Hence, the authors are advised to refine this article by giving just a short introduction of innate and adaptive immunity plus the usefulness of zebrafish to avoid boring the readers. Text between lines 176 to 194 that  introduces the development of zebrafish immune system should be reorganized to give a better understanding of development stage-dependent differentiation of macrophages and neutrophils. 

Author Response

Response to reviewers’ comments on Manuscript Biology-2127022

We thank the reviewers for their appreciation of our work and for their very constructive comments that helped us improve our manuscript. We have taken into consideration all the concerns raised by the reviewers and modified the text and figures accordingly. Please find below a detailed, point-by-point response to the reviewer’s comments.

---------------------------------------------------------------------------------------

Response to Reviewer #2's comments

Point 1: For readers who have a basic knowledge of inflammation and the advantage of using zebrafish as a research model, this article will attract the best attention if its major content begins from lines 243, as the transparency of zebrafish at embryonic stages and molecular techniques available can reveal very early events in inflammation triggered either by infection or tissue damage. Hence, the authors are advised to refine this article by giving just a short introduction of innate and adaptive immunity plus the usefulness of zebrafish to avoid boring the readers. Text between lines 176 to 194 that  introduces the development of zebrafish immune system should be reorganized to give a better understanding of development stage-dependent differentiation of macrophages and neutrophils. 

Response 1: following the reviewer’s advice we have now shortened to a minimum the part in the introduction on the advantages of the zebrafish model and the part on the description of the immune system in zebrafish (Part 2), as it is true that this has been extensively described in the literature already. Only very succinct description of the development of macrophages and neutrophils is now provided, as well as a short statement on adaptive immunity in zebrafish. The part on epithelial immunity has also been shortened. We however kept that part as a separate part in the text, for consistency, as it does not deal with molecular but with cellular actors of inflammation. In our opinion, it is also essential to keep it in the text, in order to introduce all the existing tools that allow to track and study immune cells in zebrafish and which we mention extensively in Part 4 of the manuscript. We hope this new version provide just enough information so that readers with limited knowledge in the field can still comprehend the entire review, while avoiding a long and boring description for the readers more familiar with the topic.

Reviewer 3 Report

This well-written review by Jade Leiba et al. provides a balanced overview of the cellular and molecular components and signaling pathways involved in inflammation in zebrafish, highlighting the zebrafish as a unique vertebrate model to study inflammation. The authors also summarize the tools, including the transgenic/mutant lines, morpholinos, antibodies, etc., and several means of infection with different pathogens performed in zebrafish, which would significantly benefit zebrafish researchers. The review also provides a fair representation of the field's current state with adequately cited references, which is timely and informative to the community. Specific comments were provided below on the areas that could be improved:

1.     Figure 1 is too small to see the labeling and symbols. Also, what do the arrows indicate on panel 1. initiation? I would suggest using I, II, and III as different label phases to avoid the confusion caused by labeling the inflammatory events (1-11) on the panels.

2.     Line 176-179 would be better placed in the last paragraph as it belongs to the primitive wave of hematopoiesis. 

3.     Line 550: what is a “Th2-like phenotype”? Line 880-881: what is “DSS, TNBS”?

4.      Citations of refs need to be included: several places in Tables 1 and 2; Line 642.

5.     Figure 2B: it seems there are white asterisks and arrows. What are they indicating?  2D: does the dashed line show the cutting edge of the tail fin? However, the fin tip is intact. 2E: It would be better to label the “neutrophil” on the panel. 

6.     Figure 3 is described less in the text to highlight that NFκB is the “central hub” in inflammation. Also, I would suggest labeling the four schematic panels like Fig4B-E so they can be cited precisely in the text.

7.     Please enlarge the picture of Fig4A3.

Author Response

Response to reviewers’ comments on Manuscript Biology-2127022

We thank the reviewers for their appreciation of our work and for their very constructive comments that helped us improve our manuscript. We have taken into consideration all the concerns raised by the reviewers and modified the text and figures accordingly. Please find below a detailed, point-by-point response to the reviewer’s comments.

---------------------------------------------------------------------------------------------------------------------------------------

Response to Reviewer #3's comments

Point 1: Figure 1 is too small to see the labeling and symbols. Also, what do the arrows indicate on panel 1. initiation? I would suggest using I, II, and III as different label phases to avoid the confusion caused by labeling the inflammatory events (1-11) on the panels.

Response 1: Figure 1 has been reformatted so that the three panels displaying the three phases of inflammation are now on top of each other and not next to each other. The labels and legends have also been enlarged to be more visible. These modifications should allow to have the figure displayed on a complete page and hence allow sufficient magnification for proper reading. The unfilled arrows on panel I indicate the direction of migration for immune cells, this is now stated in the figure legend. The numbering for the titles on each panel has been changed to Roman numerals to avoid confusion with the inflammatory events enumerated within the panels.

Point 2: Line 176-179 would be better placed in the last paragraph as it belongs to the primitive wave of hematopoiesis. 

Response 2: all the part describing the development of immune cells in zebrafish has now been shortened and reorganized to gain in clarity and avoid a long description that may bore readers familiar with the topic. Consequently, the development of macrophages and neutrophils is now described in a short paragraph and in chronological order, the different waves of hematopoiesis are not mentioned in detail anymore.

Point 3: Line 550: what is a “Th2-like phenotype”? Line 880-881: what is “DSS, TNBS”?

Response 3: Th2, DSS and TNBS abbreviations have now been clarified in the text, as well as a few missing others, and we have checked carefully that all abbreviations employed in the manuscript are introduced in plain when used for the first time.

Point 4: Citations of refs need to be included: several places in Tables 1 and 2; Line 642.

Response 4: Missing references have now been added in Tables 1 and 2 as well as at line 642, we thank the reviewer for pointing out that mistake.

Point 5: Figure 2B: it seems there are white asterisks and arrows. What are they indicating?  2D: does the dashed line show the cutting edge of the tail fin? However, the fin tip is intact. 2E: It would be better to label the “neutrophil” on the panel. 

Response 5: white asterisks and arrows in Figure 2B indicate activated M1-macrophages, i.e. macrophages that express TNFα. For clarity, asterisks have been replaced by white arrows and the meaning has been explained in the figure legend. On panels 2D and 2E, the dashed lines indeed represent the cutting edge for tail fin amputation. Depth of the amputation is however different in the two panels, based on the models described in the corresponding references: in Figure 2D, only the fin tip is amputated (notochord intact), while in Figure 2E, both the fin and notochord tips are amputated. This has now been specified in the legend of the figure. In the small insert in Figure 2E, “neutrophil” is now written in plain.

Point 6: Figure 3 is described less in the text to highlight that NFκB is the “central hub” in inflammation. Also, I would suggest labeling the four schematic panels like Fig4B-E so they can be cited precisely in the text.

Response 6: we have now added a full paragraph on NFκB signaling in the part on induction of inflammation, to give a few more examples that highlight the importance of this transcription factor in the inflammatory process. Together with other examples cited in the text, we believe this gives more strength to our point that NFκB-dependent pathway is a central hub in inflammation in zebrafish. Figure 3 has now been divided into 4 panels marked A-D and cited at the appropriate places in the text. The order of the panels has been changed to match the order of appearance in the text.

Point 7: Please enlarge the picture of Fig4A3.

Response 7: the scheme for static immersion on Figure 4A3 has now been enlarged for better visualization.

Round 2

Reviewer 2 Report

The text structure of introduction in Biology 2127022 has been carefully revised and the whole manuscript is now very readable. This article can be accepted for publication.